# Globally Optimal Policy Gradient Algorithms for Reinforcement Learning with PID Control Policies

**Vipul K. Sharma**
Edwardson School of Industrial Engineering
Purdue University
West Lafayette, IN 47906
`sharm697@purdue.edu`

**Wesley A. Suttle**
U.S. Army Research Laboratory
Adelphi, Maryland 20783
`wesley.a.suttle.ctr@army.mil`

**S. Sivaranjani**
Edwardson School of Industrial Engineering
Purdue University
West Lafayette, IN 47906
`sseetha@purdue.edu`

## Abstract

We develop policy gradient algorithms with global optimality and convergence guarantees for reinforcement learning (RL) with proportional-integral-derivative (PID) parameterized control policies. RL enables learning control policies through direct interaction with a system, without explicit model knowledge that is typically assumed in classical control. The PID policy architecture offers built-in structural advantages, such as superior tracking performance, elimination of steady-state errors, and robustness to model error that have made it a widely adopted paradigm in practice. Despite these advantages, the PID parameterization has received limited attention in the RL literature, and PID control designs continue to rely on heuristic tuning rules without theoretical guarantees. We address this gap by rigorously integrating PID control with RL, offering theoretical guarantees while maintaining the practical advantages that have made PID control ubiquitous in practice. Specifically, we first formulate PID control design as an optimization problem with a control policy that is parameterized by proportional, integral, and derivative components. We derive exact expressions for policy gradients in these parameters, and leverage them to develop both model-based and model-free policy gradient algorithms for PID policies. We then establish gradient dominance properties of the PID policy optimization problem, and provide theoretical guarantees on convergence and global optimality in this setting. Finally, we benchmark the performance of our algorithms on the `controlgym` suite of environments.

## 1 Introduction

Reinforcement learning (RL) for control, particularly policy gradient (PG) methods, have received significant attention in recent years (see [25] for a comprehensive survey). This has been spurred by theoretical advances in characterizing the convergence, sample complexity, and global optimality of PG methods in classical control settings like the linear quadratic regulator (LQR) [18, 28, 31], where the underlying problems have challenging, often non-convex optimization landscapes. Further developments have also extended such theoretical guarantees to additional control problems including stabilization [36], $\mathcal{H}_2$ and $\mathcal{H}_\infty$ control [48, 21], noisy LQR [23, 20], output-feedback [17] and partially observed problems like linear quadratic Gaussian (LQG) control [44, 16, 30], nonlinear control [42], and robust control problems including adversarial disturbances [47, 49].

39th Conference on Neural Information Processing Systems (NeurIPS 2025).

The proportional-integral-derivative (PID) control architecture has received surprisingly limited attention within the RL for control literature, despite its prominent place within control theory and engineering practice [2, 4]. PID control offers built-in structural advantages due to its control law being parameterized by three interpretable components: proportional action to shape stability and transient performance, integral action to eliminate tracking errors, and derivative action based on anticipated future error to enhance robustness. Due to these properties, PID control is a widely adopted paradigm in several domains including industrial control, aerospace, and robotics applications [10], with over 90% controllers used in industry employing PID architectures [22]. Further, the output-feedback architecture and low-dimensional parameterization of PID control make it a promising avenue for policy-gradient based exploration. However, most PID control design approaches remain heuristic, relying on engineering intuition and lacking provable guarantees on performance, optimality, or even stability [4, 40].

In this paper, we bridge this gap by rigorously integrating PG approaches from RL with the PID control parameterization, enabling us to obtain convergence and global optimality guarantees enjoyed by PG methods for control while maintaining the practical advantages that have made PID control ubiquitous in practice. Specifically, we first provide an optimization-based formulation of the PID control problem, and derive exact policy gradient expressions in the PID parameters. Based on these gradient expressions, we develop two new PG algorithms, a model-based approach for PID control and a model-free approach for PI control. We prove that the objective of these problems enjoys the gradient dominance property, establish convergence rates to global optimality for our model-based approach, and characterize the sample complexity needed to achieve approximate global optimality for our model-free algorithm. Finally, we benchmark the performance of our algorithms on the `controlgym` suite of control environments [51]. We summarize our core contributions as follows:

- **Policy optimization framework and algorithms.** Our optimization-based formulation of the PID problem and our proof that its objective enjoys gradient dominance together establish a critical link between the PID control problem and the literature on PG methods for control. Our derivation of exact gradient expressions for this problem also paves the way for the development of principled PG algorithms for tuning PID control policies. Specifically, we develop two new PG algorithms: a model-based method leveraging system identification to learn PID control policies, and a model-free method for learning PI control policies. In the model-free setting, we bypass the need for model knowledge by applying the classic policy gradient theorem, and derive a novel algorithm that employs stochastic policies during training to provably learn the optimal parameters of an underlying deterministic PI control policy. This allows us to provably achieve approximate global optimality in a model-free manner on PI problems, resolving an important open problem.

- **Global optimality, convergence, sample complexity.** We establish global optimality, convergence, and sample complexity guarantees for our algorithms. For our model-based approach, we leverage gradient dominance to establish a linear convergence rate to global optimality. In the model-free setting, we prove that our optimization objective enjoys weak gradient dominance and establish convergence rate and sample complexity guarantees to global optimality. In the model-free setting, the analysis is particularly challenging due to the use of stochastic policies to learn the parameters of an underlying deterministic policy. To overcome this, we establish weak gradient dominance under stochastic policies, then use this to derive the algorithm's sample complexity for achieving approximate global optimality. While there are some recent results [33] on theoretical guarantees for learning deterministic policies via stochastic gradients, these analyses assume global Lipschitz properties, which are not satisfied by PID control policies. In contrast, we obtain theoretical guarantees of convergence and sample complexity for a class of problems that only satisfy less restrictive local Lipschitz properties. In addition, our model-free approach yields a first-order PG algorithm that is applicable not only to PI control but also to general LQR problems (which constitute a special case of our framework where the control policy is parameterized only by the proportional component of the PID policy), whereas only zero-order methods (where gradients are estimated using only objective function evaluations) were previously available [18].

## 1.1 Related Work

While to our knowledge there is no literature integrating policy optimization theory with PID control, there is a rich literature on PG methods for problems like LQR, as well as attempts from control theory to introduce optimization objectives into PID control, which we review here.

**Policy gradient methods.** Convergence of PG methods to global optimality, often referred to as *global convergence* in the machine learning community, has received significant attention in recent years, starting with the fundamental results of [8, 29, 1, 26]. Subsequent works addressed optimal convergence rates [32], sample complexity [7], regret [6, 35], and even global convergence for general utilities [50, 41] and deterministic policies [33]. A key feature of these analyses lies in establishing gradient dominance of the optimization objective, which allows global optimality to be established despite the typical non-convexity of the problem. Within RL for control, PG methods for a variety of control problems [25] have been shown to enjoy gradient dominance, particularly LQR and its variants where model-based and zero-order model-free PG methods have been developed [18, 28], PG for least squares problems has been analyzed [13], sample complexity results for PG methods have been provided [14], and system identification-based approaches developed [46]. Despite the wide range of works on PG for control, to our knowledge, globally convergent PG methods for PID control remain unexplored in the literature. We close this gap in this paper.

**Optimization in PID Control.** While most PID designs rely in practice on heuristic tuning rules and simplified models, there have been attempts to formalize PID control design as an optimization problem [3, 5, 24]. Constrained optimization approaches for PID control with robustness specifications [34, 19] and linear and bilinear matrix inequality-based tuning for single and multi-variable settings have also been proposed [11, 45, 9]. However, all of these methods rely on frequency domain system descriptions, making them difficult to integrate into RL theory, where state space descriptions are fundamental. Critically, these approaches lack theoretical guarantees on optimality and convergence. Recently, [37] proposed a new optimization framework connecting model-based PID control to the standard continuous-time LQR problem. We adopt a similar approach to formulate an optimization problem for PID control design, albeit in the discrete-time setting. Importantly, our work goes far beyond the scope of [37] by providing exact policy gradient expressions and rigorous theoretical guarantees in both the model-based and model-free settings.

## 2 Problem Formulation

We focus on the following optimal control problem with linear system dynamics and proportional-integral-derivative (PID) control policies, given by

$$\text{minimize} \quad \mathbb{E}\left[\sum_{t=0}^{\infty} y_t^T Y y_t + u_t^T R u_t\right] \tag{1}$$
$$\text{subject to} \quad x_{t+1} = Ax_t + Bu_t, \quad y_t = Cx_t, \quad x_0 \sim \mathcal{D},$$

where $x_t \in \mathbb{R}^n$ is the system state at time $t \geq 0$, $y_t \in \mathbb{R}^p$ represents the output, and $u_t \in \mathbb{R}^m$ denotes the output of the control policy. We henceforth adopt the scalar input and output setting $(m, p = 1)$ for ease of exposition, but we note that all of our theory can be extended to the multivariable setting. The control policy is parameterized by the PID parameters $\tilde{K} = [K_P, K_I, K_D]^T$, where $K_P, K_I, K_D \in \mathbb{R}$, as

$$u_t = -K_P y_t - K_D \frac{y_{t+1} - y_t}{\tau} - K_I \sum_{j=0}^{k-1} y_j, \tag{2}$$

and $\tau > 0$ is the sampling time of the discrete-time dynamics. The system is initialized at $x_0 \sim \mathcal{D}$ according to some probability distribution $\mathcal{D}$ over the state space, $Y \in \mathbb{R}^{p \times p}$ and $R \in \mathbb{R}^{m \times m}$ are positive definite matrices, and $A \in \mathbb{R}^{n \times n}$ and $B \in \mathbb{R}^{n \times m}$ are the system and the input matrices. Intuitively, for a classical tracking problem, where the objective is to drive the system state to track an external reference, the proportional parameter $K_P$ responds to the current tracking error, the integral parameter $K_I$ addresses long-term errors by accumulating the error signal over time, and the derivative parameter $K_D$ predicts future errors by evaluating the rate of change of the error. The PID control problem presented in equations (1)-(2) in the PID parameters $\tilde{K}$ is equivalent to a related problem, as formalized in the following proposition. We defer all proofs to the appendix.

**Proposition 2.1.** *The optimization problem* (1) *is equivalent to the following problem:*

$$\text{minimize} \quad J(\tilde{K}) = \mathbb{E}\left[\sum_{t=0}^{\infty} g_t^T Q g_t + u_t^T R u_t\right] \tag{3}$$
$$\text{subject to} \quad g_{t+1} = \bar{A}x_t + \bar{B}u_t, \quad g_0 \sim \mathcal{D},$$

*where*

$$g_t := \begin{bmatrix} x_t \\ z_t \end{bmatrix}, \qquad z_t := \sum_{i=0}^{t-1} y_i = \sum_{i=1}^{t-1} Cx_t, \qquad z_0 = 0 \tag{4}$$

$\bar{A} = \begin{bmatrix} A & 0 \\ C & I \end{bmatrix}, \bar{B} = \begin{bmatrix} B \\ 0 \end{bmatrix}, Q = \begin{bmatrix} C^T Y C & 0 \\ 0 & 0 \end{bmatrix}$, *Y is as in* (1), *and* $g_0 \sim \mathcal{D}$ *is used to signify* $x_0 \sim \mathcal{D}, z_0 = 0$ *with a slight abuse of notation. The control policy in* (3) *is given by*

$$u_t = -Kg_t, \tag{5}$$

*where the control parameter K is obtained from the PID parameters* $\tilde{K} = [K_P, K_I, K_D]^T$ *as*

$$K = \phi(\tilde{K}) = \left[1 + \frac{K_D CB}{\tau}\right]^{-1} \left[K_P C + K_D \frac{C(A-I)}{\tau} \quad K_I\right]. \tag{6}$$

Proposition 2.1 extends the continuous-time results in [37] to our discrete-time PID setting, establishing an equivalence between the PID control problem and an LQR problem in an extended state space. By (6), the parameter $K$ is completely determined given the PID parameters $\tilde{K}$. For this reason, in what follows we will abuse notation and use $J(K)$ and $J(\tilde{K})$ interchangeably, depending on context. Specifically, when we write $J(\tilde{K})$ and $\nabla_{\tilde{K}} J(\tilde{K})$, we consider $J(\phi(\tilde{K}))$ and $\nabla_{\tilde{K}} J(\phi(\tilde{K}))$, respectively. It is important to note that PG methods for LQR (e.g., [18]) do not apply directly to the PID control problem, since, though $K$ in (5) is completely determined given the PID parameters $\tilde{K}$ by (6), there does not necessarily exist a feasible decomposition of $K$ in terms of $\tilde{K}$. For this reason, it is not possible to use existing PG methods for LQR to solve (3), then translate the resulting policy into a feasible PID policy. Fortunately, it is possible to perform PG descent directly in the PID parameters. To achieve this, we derive gradient expressions in the PID parameters in the following section.

## 3 Algorithms

In this section we develop PG algorithms for optimal tuning of PID and PI controllers. In order to achieve this, we first derive policy gradient expressions for the objective (3) with respect to the PID parameters in Theorem 3.1, then build on this result to derive our algorithms. The first algorithm is a model-based approach that leverages system identification [27] to learn a controller for the full PID problem. The second algorithm is a model-free approach for learning a PI controller. We assume throughout that the matrix $C$ appearing in (1) is known, which is common in practice, since the output $y$ is typically designed depending on the application at hand.

**PID Gradient Expressions.** We begin by deriving expressions for the gradients of $J(\tilde{K})$ with respect to $K_P, K_I, K_D$. We will use these expressions in the remainder of this section to develop PG algorithms for performing gradient descent in the PID parameters. In order to obtain closed-form gradient expressions for $J(\tilde{K})$ in terms of the PID parameters $\tilde{K}$, we first develop additional notation concerning the closed-loop dynamics of (3). First, from $g_{t+1} = \bar{A}g_t + \bar{B}u_t$ and $u_t = -Kg_t$, we get

$$g_{t+1} = A_K g_t, \qquad A_K = \bar{A} - \bar{B}K. \tag{7}$$

Expanding the definitions of $\bar{A}, \bar{B}$ yields

$$A_K = \begin{bmatrix} A - BF_D K_P C - BF_D K_D \frac{C(A-I)}{\tau} & -BF_D K_I \\ C & I \end{bmatrix}, \tag{8}$$

where

$$F_D = [1 + \frac{K_D CB}{\tau}]^{-1}. \tag{9}$$

For this closed-loop system, the corresponding cost function is given by $J(\tilde{K}) = \mathbb{E}\left[g_0^T P_K g_0\right]$, where $P_K$ is the solution to the discrete algebraic Riccati equation

$$A_K^T P_K A_K - P_K + Q + K^T RK = 0, \tag{10}$$

and $K$ is obtained from $\tilde{K}$ via (6). We now provide gradient expressions for the objective of (3) in the PID parameters.

**Algorithm 1** PG4PID

1: **Input:** Tolerance $\varepsilon$, stepsizes $\alpha_P, \alpha_I, \alpha_D$, trajectory length $N$
2: **Initialize:** $K_P^0, K_I^0, K_D^0$, set $t \leftarrow 0$
3: Generate $x_0, u_0, \ldots, x_N, u_N$ with (5)
4: Form $X_N, U_N$ from the above trajectory
5: Estimate $A, B$ from $X_N, U_N$ using (77)
6: **repeat**
7:   Compute $K_t$ from $\tilde{K}_t = [K_P^t, K_I^t, K_D^t]^T$ using (6)
8:   Estimate $P_{K_t}$ using (88)
9:   Compute $E_{K_t}$ using (14)
10:   Compute $\nabla_{K_P} J(\tilde{K}_t), \nabla_{K_I} J(\tilde{K}_t), \nabla_{K_D} J(\tilde{K}_t)$ from Theorem 3.1
11:   $K_P^{t+1} \leftarrow K_P^t - \alpha_P \nabla_{K_P} J(\tilde{K}_t)$
12:   $K_I^{t+1} \leftarrow K_I^t - \alpha_I \nabla_{K_I} J(\tilde{K}_t)$
13:   $K_D^{t+1} \leftarrow K_D^t - \alpha_D \nabla_{K_D} J(\tilde{K}_t)$
14:   $t \leftarrow t + 1$
15: **until** $\left\| \nabla_{\tilde{K}} J(\tilde{K}_t) \right\| < \varepsilon$

**Algorithm 2** PG4PI

1: **Input:** Tolerance $\varepsilon$, stepsizes $\alpha_P, \alpha_I$, trajectory length $N$, noise parameter $\sigma > 0$
2: **Initialize:** $\tilde{K}_0 := [K_P^0 \ K_I^0]^T$, set $t \leftarrow 0$
3: **repeat**
4:   Sample $g_0 \sim \mathcal{D}$
5:   Generate $g_0, u_0, c_0, \ldots, g_{N-1}, u_{N-1}, c_{N-1}$ with $\pi_{\tilde{K}}$
6:   $C_t \leftarrow \sum_{k=0}^{N-1} c_k$
7:   Compute $\nabla_{K_i} \log \pi_{\tilde{K}}(u_0 | g_0), i = P, I$ using (22), (23)
8:   $K_P^{t+1} \leftarrow K_P^t - \alpha_P C_t \nabla_{K_P} \log \pi_{\tilde{K}}(u_0 | g_0)$
9:   $K_I^{t+1} \leftarrow K_I^t - \alpha_I C_t \nabla_{K_I} \log \pi_{\tilde{K}}(u_0 | g_0)$
10:   $t \leftarrow t + 1$
11: **until** $\left\| \nabla_{\tilde{K}} J(\tilde{K}_t) \right\| < \varepsilon$

**Theorem 3.1.** *The gradients of the objective $J(\tilde{K})$ of (3) with respect to $K_P, K_I, K_D$ are given by*

$$\nabla_{K_P} J(\tilde{K}) = 2F_D E_K \Sigma_K T_x^T C^T, \tag{11}$$

$$\nabla_{K_I} J(\tilde{K}) = 2F_D E_K \Sigma_K T_z^T, \tag{12}$$

$$\nabla_{K_D} J(\tilde{K}) = 2F_D E_K \Sigma_K (\alpha_1 T_x^T (A - I)^T C^T - \alpha_2 \alpha_3), \tag{13}$$

*where* $T_x = [I_n \ 0_{n \times 1}], T_z = [0_{1 \times n} \ 1], \alpha_1 = (\tau + K_D C B)^{-1}, \alpha_2 = (\tau + K_D C B)^{-1} (\tau T_x^T C^T K_P + T_x^T (A - I)^T C^T K_D + \tau T_z^T K_D), \alpha_3 = C B,$ *and*

$$E_K = (R + \bar{B}^T P_K \bar{B}) K - \bar{B}^T P_K \bar{A}, \quad (14) \qquad \Sigma_K = \mathbb{E} \sum_{t=0}^{\infty} g_t g_t^T. \tag{15}$$

Recall that $E_K$ and $\Sigma_K$ are functions of $K$ and therefore of $\tilde{K}$ by (6). Equipped with the explicit gradient expressions provided by Theorem 3.1, we turn next to developing algorithms using them.

## 3.1 Model-based Policy Gradient

We now derive our model-based PG algorithm for solving (1). Our approach is model-based in the sense that it relies on explicitly estimating the system matrices $A$ and $B$ and the solution $P_K$ to the Riccati equation (10) from collected trajectory data. Our algorithm builds on the approach proposed in [46], with the key difference that our procedure for gradient estimation procedure requires estimation of fewer system matrices, thereby enjoying superior computational efficiency.

To estimate $A$ and $B$, we follow a least-squares system identification approach similar to that proposed in [27]. Given trajectory data $x_0, u_0, x_1, u_1, \ldots, x_{t+1}, u_{t+1}$ collected from the system, for some $t > 0$, and taking $k \le t$, let $X_k = [x_0 \ x_1 \ \ldots \ x_{k-1}], U_k = [u_0 \ u_1 \ \ldots \ u_{k-1}]$ denote matrices whose columns are formed, respectively, by the states $x_0, \ldots, x_{k-1}$ and inputs $u_0, \ldots, u_{k-1}$. Using this trajectory data, we can estimate $A$ and $B$ by simply computing the least-squares estimator of the error $\|X_{t+1} - (AX_t + BU_t)\|^2$ in terms of $A$ and $B$, as described in Section A.2.2 in the Appendix. To estimate $P_{K_t}$ for $K_t$ at time $t \ge 0$, we use the iterative Riccati equation solution approach described in Section A.2.2 in the appendix. Combining the above procedures for estimating $A, B$, and $P_K$ with the gradient expressions of Theorem 3.1, we provide the model-based PG algorithm in Algorithm 1.

## 3.2 Model-free Policy Gradient

Though the model-based Algorithm 1 achieves a globally optimal PID controller as shown in Section 4, direct estimation of $A, B$, and $P_K$ can be computationally burdensome in practice. To address this, we now propose a model-free PG approach to learn globally optimal PI controllers without the need to estimate the system dynamics. The key to our approach is to use a stochastic policy centered around the deterministic PID policy (5), whose form allows us to apply the PG theorem [43] to perform model-free learning of optimal PI control policies. While the use of Gaussian policies is widespread in the RL literature, the key innovation in our work is the use of the PID parameterization (5) for the mean. This allows us to combine the PG expressions of Theorem 3.1 with the PG theorem [43] to directly tune the parameters of our control policy in a model-free manner.

For a state $g$ in the augmented state space of problem (3), consider the deterministic policy $\mu_{\tilde{K}}(g) = -Kg$ of (5). We define a stochastic policy based on this deterministic controller by

$$\pi_{\tilde{K}}(u|g) = \frac{1}{\sigma\sqrt{2\pi}}e^{-\frac{(u-\mu_{\tilde{K}}(g))^T(u-\mu_{\tilde{K}}(g))}{2\sigma^2}}, \tag{16}$$

where $\sigma > 0$ is a user-specified constant. Notice that (16) is simply the probability density function corresponding to $\mathcal{N}(\mu_{\tilde{K}}(g), \sigma^2 I)$, a Gaussian random variable with mean $\mu_{\tilde{K}}(g)$ and covariance $\sigma^2 I$, where $I$ is the identity matrix. Sampling a control input $u \sim \pi_{\tilde{K}}(\cdot|g)$ is thus equivalent to using the control $\mu_{\tilde{K}}(g)$ plus zero-mean Gaussian noise. Since the policy in (16) is stochastic, it can be used in the policy gradient theorem, which states in the finite-horizon case that, for initial $g_0$ and horizon $N$,

$$\nabla J(\tilde{K}) = \mathbb{E}\Big[\sum_{t=0}^{N-1} c(g_t, u_t)\nabla \log \pi_{\tilde{K}}(u_0|g_0)\Big], \tag{17}$$

where $c(g_t, u_t) := g_t^T Q g_t + u_t^T R u_t$. This expression enables truly model-free approaches to solving (3), since it provides a way to minimize costs given access only to information about costs and the policy $\pi_{\tilde{K}}$. Note that, in order for (17) to be well-defined, it is critical that $\pi_{\tilde{K}}(u|g) > 0$, for all $g, u$. The stochastic policy $\pi(\cdot|g)$ satisfies this property, while the deterministic controller $\mu_{\tilde{K}}(g)$, which can be viewed as the limiting case of (16) as $\sigma \to 0$, does not. This highlights the necessity of introducing the stochastic policy (16) in order to apply (17).

Notice that (17) holds whether the gradient is taken with respect to $K$ or $\tilde{K}$, again by equation (6). In our setting, if the score function $\nabla_{K_i} \log \pi_{\tilde{K}}(u_0|g_0)$ is well-defined and efficiently computable, for each $i = P, I, D$, then (17) can be used to approximate $\nabla_{K_i} J(\tilde{K})$ by sampling $g_0 \sim \mathcal{D}, u_0 \sim \pi_{\tilde{K}}(\cdot|g_0)$, then collecting an $N$-length trajectory $g_0, u_0, g_1, u_1, \ldots, g_{N-1}, u_{N-1}$, where $N$ is sufficiently large. We therefore need expressions for computing $\nabla_{K_i} \log \pi_{\tilde{K}}(u|g)$, for $i = P, I, D$. First notice that, for $i = P, I, D$, equation (16) and the chain rule give

$$\nabla_{K_i} \log \pi_{\tilde{K}}(u|g) = \frac{1}{\sigma^2}(u - \mu_{\tilde{K}}(g))\nabla_{K_i}\mu_{\tilde{K}}(g). \tag{18}$$

Thus, to apply (17), all we need are the following expressions for $\nabla_{K_i}\mu_{\tilde{K}}(g), i = P, I, D$.

**Theorem 3.2.** *For $K$ defined in* (6)*, we have*

$$\nabla_{K_P}\mu_{\tilde{K}}(g) = -\tau(\tau + K_D CB)^{-1}CT_x g \tag{19}$$

$$\nabla_{K_I}\mu_{\tilde{K}}(g) = -\tau(\tau + K_D CB)^{-1}T_z g \tag{20}$$

$$\nabla_{K_D}\mu_{\tilde{K}}(g) = -(\tau + K_D CB)^{-1}(C(A - I)T_x g - KgCB). \tag{21}$$

It is clear from Theorem 3.2 that $\nabla_{K_P}\mu_{\tilde{K}}(g)$ and $\nabla_{K_I}\mu_{\tilde{K}}(g)$ depend on $B$, while $\nabla_{K_D}\mu_{\tilde{K}}(g)$ depends on both $A$ and $B$. Due to the presence of $K_D$ in the denominator, however, we note that fixing $K_D = 0$ eliminates this dependence on $A$ and $B$ in (19)-(20), allowing us to estimate gradients for PI policies in a truly model-free way. Combining (18) with Theorem 3.2, we immediately have:

**Corollary 3.3.** *For PI control, with $K_D = 0$, we have*

$$\nabla_{K_P} \log \pi_{\tilde{K}}(u|g) = \frac{g^T T_x^T C^T(\mu_{\tilde{K}}(g) - u)}{\sigma^2}, \quad (22) \qquad \nabla_{K_I} \log \pi_{\tilde{K}}(u|g) = \frac{g^T T_z^T(\mu_{\tilde{K}}(g) - u)}{\sigma^2}. \quad (23)$$

In the convergence results of the following section, $\sigma$ in (22)-(23) is chosen to be small enough that suitable gradient dominance and convergence results can still be recovered. In practice, the choice of

$\sigma$ comes down to ensuring numerical stability, and in our experiments we observed that Algorithm 2 remained stable for a wide variety of $\sigma$ values. Pseudocode for our model-free PI approach based on Corollary 3.3 is provided in Algorithm 2. Though our model-free algorithm provides only PI instead of PID control, we emphasize that it is still broadly applicable, since active tuning of the derivative component of PID controllers is uncommon in practice [4].

## 4 Convergence Analysis

In this section, we provide convergence and global optimality guarantees for Algorithms 1 and 2. We first provide a detailed overview of our results. Denote the expected costs under the deterministic policy (5) and the stochastic, Gaussian policy (16), respectively, by

$$ J_\mu^D(\tilde{K}) = \mathbb{E}_{\mu_{\tilde{K}}} \Big[ \sum_{t=0}^\infty y_t^T Y y_t + u_t^T R u_t \Big], \quad (24) \qquad J_\pi^G(\tilde{K}) = \mathbb{E}_{\pi_{\tilde{K}}} \Big[ \sum_{t=0}^\infty y_t^T Y y_t + u_t^T R u_t \Big], \quad (25) $$

and let $\tilde{K}^*$ and $\tilde{K}_G^*$ denote the global minimizers of (24) and (25), respectively. Note that, due to the structure of problem (3), such optimal parameters are guaranteed to exist and to be unique [18]. In what follows, we first establish that the objective (24) enjoys the gradient dominance property in the PID parameters $\tilde{K}$, which implies that any first-order stationary point of (24) is globally optimal. This key result, which leverages Proposition 2.1 and our PID policy gradient results of Theorem 3.1, lays the foundation for the global optimality guarantees for both algorithms. Second, we establish the convergence rate of Algorithm 1 to the global minimizer $\tilde{K}^*$ of (24). Finally, we provide convergence rate and sample complexity results demonstrating that Algorithm 2 achieves an approximately globally optimal solution of (24). We highlight that existing analyses of PG for LQR [18] and global optimality of stochastic PG methods for learning deterministic policies [33] require gradient dominance and globally Lipschitz system dynamics, whereas we improve on these results in our analysis by requiring only weak gradient dominance [32] and locally Lipschitz system dynamics.

**PID Gradient Dominance.** We now establish gradient dominance of the objective (3) in the PID parameters $\tilde{K}$, which implies that any algorithm that achieves a first-order stationary point of the problem in fact achieves global optimality. For any matrix $M$, let $\rho(M)$ and $\sigma_{min}(M)$ denote its spectral radius and minimal singular value, respectively, and let $M_{PID} = \big[ (\nabla_K K_P)^T \ (\nabla_K K_I)^T \ (\nabla_K K_D)^T \big]^T$. Finally, define the set of stabilizing PID parameters as

$$ \tilde{\mathcal{K}} := \{ (K_P, K_I, K_D) \in \mathbb{R}^3 : \rho(\bar{A} - \bar{B}K) < 1, K \text{ satisfies (6)} \}. \quad (26) $$

We have the following result.

**Theorem 4.1.** *Let $\tilde{K}^* = \min_{\tilde{K} \in \tilde{\mathcal{K}}} J(\tilde{K})$. Then*

$$ \Big| J(\tilde{K}) - J(\tilde{K}^*) \Big| \le \tilde{\alpha} \left\| \nabla_{\tilde{K}} J(\tilde{K}) \right\|^2, \quad \text{for all } \tilde{K} \in \tilde{\mathcal{K}}, \quad (27) $$

*where $\tilde{\alpha} = \| \Sigma_{K^*} \| \left[ \sigma_{min}(\Sigma_K)^2 R \right]^{-1} \| M_{PID} \|^2$, $K^*$ is obtained using (6) with $\tilde{K} = \tilde{K}^*$, and, $\Sigma_{K^*}$ is obtained from (15) with $K = K^*$.*

As mentioned above, Theorem 4.1 implies that any first-order stationary point of $J(\tilde{K})$ is in fact globally optimal, i.e., that if $\tilde{K}'$ satisfies $\nabla_{\tilde{K}} J(\tilde{K}') = 0$, then $\tilde{K}'$ is optimal, since $J(\tilde{K}') = J(\tilde{K}^*)$. For this reason, the development of PG algorithms for (3) is well-motivated: PG algorithms performing gradient descent in $\tilde{K}$ on this problem can be reasonably expected to achieve global optimality. We next leverage Theorem 4.1 to show that Algorithms 1 and 2 do precisely this.

### 4.1 Model-based policy gradient for PID

We now establish the convergence rate of Algorithm 1 to the global minimizer $\tilde{K}^*$ of (24). As we will see, Algorithm 1 achieves global optimality due to two factors: (i) the algorithm directly learns the parameters of the deterministic controller (5), and (ii) the problem (3) enjoys gradient dominance in the PID parameters, as established in Theorem 4.1 above. The following result, which partially extends the result [13, Theorem 3] to the PID setting, establishes the convergence rate of Algorithm 1 to the globally optimal objective function value.

**Theorem 4.2.** *Let $\eta > 0$ be as in Section A.3.1. Fix $\eta' \in (0, \eta)$, let Algorithm 1 be run with stepsizes $\alpha_P, \alpha_I, \alpha_D = \eta'$, and let $\{\tilde{K}_t\}$ denote the sequence of parameters generated. Then there exist $\kappa \in (0, 1)$ such that*

$$J_\mu^D(\tilde{K}_t) - J_\mu^D(\tilde{K}^*) \le \kappa^t \left( J_\mu^D(\tilde{K}_0) - J_\mu^D(\tilde{K}^*) \right). \tag{28}$$

This theorem establishes linear convergence of Algorithm 1 to a globally optimal solution in the case where $A, B$ and $P_{K_t}$ are estimated perfectly at each step. This result can be generalized to incorporate estimation error in the system identification procedure by following analysis techniques such as those in [46]. Since we focus on the development of globally convergent PG methods for PID control in this paper, we leave this for future work.

### 4.2 Model-free policy gradient for PI

We now turn to providing convergence and global optimality guarantees for Algorithm 2. Specifically, we establish convergence rate and sample complexity results for achieving within $\beta > 0$ of an $\epsilon$-global solution of (24), where $\beta$ is controlled by choice of variance $\sigma^2$ and rollout length $N$ in Algorithm 2. It is important to observe that, despite the fact that Algorithm 2 actually optimizes (25), it is nonetheless able to achieve near-optimality for (24) for appropriate choice of $\sigma$ and $N$. This is because the objective (25) enjoys weak gradient dominance, as we prove in Theorem 4.4. Similar to [33], the reason that approximate instead of exact $\epsilon$-optimality is achieved is due to the use of the stochastic version (16) of the deterministic controller (5).

We now establish that the objective (25) of Algorithm 2 enjoys the weak gradient dominance property studied in [33], which is critical to establishing the global optimality guarantee below. We prove this result under the following mild assumption, which is a technical condition ensuring that the difference between the costs (24) and (25) remains bounded.

**Assumption 4.3.** The variance of the stochastic policy (16) at time $t \ge 0$ is $\sigma_t^2 = \sigma^2/(1+t)^2$.

Assumption 4.3 is akin to technical assumptions on decreasing step size that are common across most gradient-based algorithms, including stochastic gradient descent (SGD). This assumption can likely be relaxed in practice, since we observe across all our experiments in Section 5 that PG4PI achieves convergence even with a constant $\sigma_t$, that is $\sigma_t = \sigma, \forall t \ge 0$. In addition, our experimental ablation studies in A.4.2 also demonstrate convergence for a wide range of fixed values of $\sigma$.

**Theorem 4.4.** *Let Assumption 4.3 hold and let $\tilde{K}_G^*$ be a minimizer of (25). We have*

$$\left| J_\pi^G(\tilde{K}) - J_\pi^G(\tilde{K}_G^*) \right| \le \left\| \tilde{\alpha} \nabla_{\tilde{K}} J_\pi^G(\tilde{K}) \right\|^2 + \beta, \tag{29}$$

*for all $\tilde{K} \in \tilde{\mathcal{K}}$, where $\tilde{\mathcal{K}}$ is as in (26) and $\beta = \frac{2\|R\|\sigma^2\pi^2}{3}$.*

With Theorem 4.4 in hand, we turn to our main result. Denote the estimate obtained from (17) by

$$\widehat{\nabla_{\tilde{K}} J_\pi^G}(\tilde{K}) = \sum_{t=0}^{N-1} c(g_t, u_t) \nabla_{\tilde{K}} \log \pi_{\tilde{K}}(u_0|g_0). \tag{30}$$

We will need the following conditions on (30).

**Assumption 4.5.** There exists finite $\bar{V} > 0$ such that, for all $\tilde{K}, g_0, u_0$ and $N \in \mathbb{N}$, we have $\mathrm{Var}\|\widehat{\nabla_{\tilde{K}} J_\pi^G}(\tilde{K})\| \le \bar{V}/N$.

**Assumption 4.6.** For $\{\tilde{K}_t\}$ generated by Algorithm 2 we have $\|\widehat{\nabla_{\tilde{K}} J_\pi^G}(\tilde{K}_t)\| \le 1$, for all $t \le T$.

Assumption 4.5 is a mild condition also assumed in [33], and can likely be shown to hold under reasonable conditions on $\sigma$ and mild continuity and differentiability properties of $J_\pi^G$. In practice, we can enforce this assumption by choosing a sufficiently long rollout length, which is a common strategy to decrease variance of PG estimates in RL. Assumption 4.6 is for technical convenience in Corollary A.9's proof and can be dispensed with by using appropriate gradient normalization in Algorithm 2. We observed in our experiments in Section 5 that the proposed algorithm PG4PI achieves convergence even without such gradient normalization, indicating that Assumption 4.6 can be relaxed in practice. We are now ready to provide our sample complexity result for Algorithm 2.

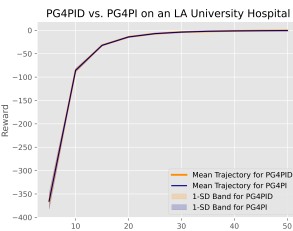 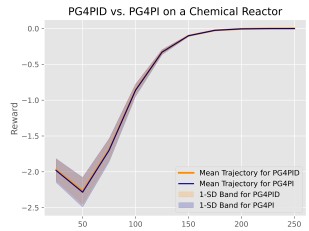 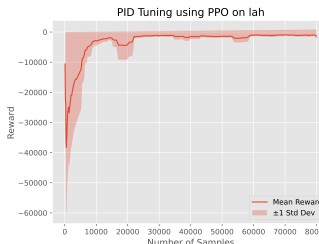

Figure 1: Reward curves for Algorithm 1 (PG4PID), Algorithm 2 (PG4PI), and PPO-based PID on the 8-dimensional Chemical Reactor and 48-dimensional LA University Hospital environments. Plots show mean and one standard deviation over multiple independent training runs.

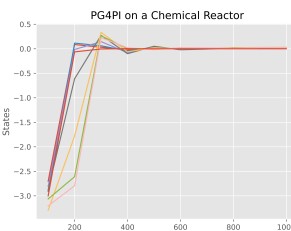 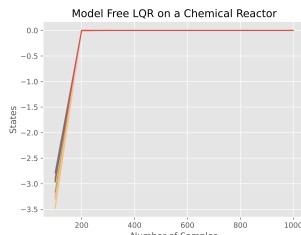 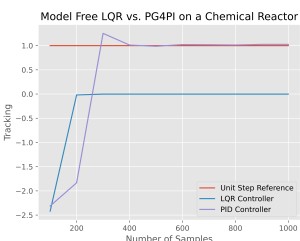

Figure 2: Stabilization and reference tracking curves for Algorithm 2 (PG4PI) and model-free LQR, on the 8-dimensional Chemical Reactor environment. Plots show that both PG4PI and LQR stabilize the states to origin; however, LQR fails to track the unit-step reference input in contrast to the perfect tracking obtained with PG4PI.

**Theorem 4.7.** *Let Assumptions 4.3, 4.5, and 4.6 hold. Let Algorithm 2 run for $T > 0$ iterations with rollout length $N > 0$ and stepsizes $\alpha_P = \alpha_I = \eta$ satisfying $\eta \leq \min\{\frac{1}{l_{mx}^T}, \frac{\tilde{\alpha}^2}{c_0}, \frac{N\tilde{\alpha}^2}{l_{mx}^T \bar{V}}\}$. Let $\{\tilde{K}_t\}_{t=0,\dots,T}$ denote the sequence of iterates generated and $\beta = 2\|R\|\sigma^2\pi^2$. Then, we have*

$$\mathbb{E}\left[J_\pi^G(\tilde{K}_T)\right] - J_\mu^D(\tilde{K}^*) \leq \beta + c_T \left(1 - \sqrt{\frac{\eta^3 l_{mx}^T \bar{V}}{4\tilde{\alpha}^2 N}}\right)^T + \sqrt{\frac{l_{mx}^T \bar{V} \eta \tilde{\alpha}^2}{N}}, \tag{31}$$

*where $c_k := \max\{0, J_\pi(\tilde{K}_k) - J_\pi^G(\tilde{K}^*) - \beta/3\}$ for $0 \leq k \leq T$, and, $l_{mx}^T = \max\{l(\tilde{K}_0), \dots, l(\tilde{K}_T)\}$, where $l(\tilde{K})$ is the local Lipschitz constant defined in Corollary A.8.*

*Moreover, to guarantee $\mathbb{E}[J_\pi^G(\tilde{K}_T)] - J_\mu^D(\tilde{K}^*) \leq \epsilon + \beta$, for $\epsilon > 0$, setting the stepsize to be $\eta = \frac{(\epsilon\tilde{\alpha})^2 N}{4 l_{mx}^T \bar{V}}$ yields sample complexity at most*

$$NT = \frac{16\tilde{\alpha}^4 l_{mx}^T \bar{V}}{\epsilon^3} \log\left(\frac{c_T}{\epsilon}\right). \tag{32}$$

Theorem 4.7 provides a useful tool for striking a balance between suboptimality and sample complexity: choosing $\sigma$ sufficiently small and $N$ sufficiently large improves the convergence rate and suboptimality as quantified in (31), while (32) quantifies the corresponding increase in sample complexity. We emphasize that [33] assumes the objective and its gradient are globally Lipschitz, while the objective of (3) is known to only satisfy local Lipschitz properties [18, 28]. To overcome this, we quantify the maximum possible deviation that can occur given the local Lipschitz properties over the entire sequence of parameter iterates, then incorporate this into the bounds (31)-(32). See Appendix A.3 for details.

# 5 Experiments

**Experiment setup.** We evaluate the performance of Algorithms 1 and 2 and compare with baseline methods on two environments from the `controlgym` library [51]: the 8-dimensional state-space Chemical Reactor and a 48-dimensional state-space LA University Hospital environments. Both environments are tracking problems, where the objective is to stabilize the system and ensure that the output tracks the reference input. On each environment we ran independent replications (see appendix for details) of Algorithms 1 and 2, the PG method for LQR from [18], and the `StableBaselines3` [38] implementation of PPO [39] with a policy parameterization mapping states to PID parameters. We collected rewards (i.e., negative costs), tracking error, time-domain tracking information, and state evolutions encountered during training. For our figures, we plotted reward, tracking performance, tracking error, and state evolution as a function of the number of samples used, i.e., the number of interactions the algorithm performed in the environment during training. We provide experiments illustrating reward performance for Algorithms 1, 2, and PPO in Fig. 1, tracking performance and error in Appendix A.4.1, ablation studies comparing various values of $\sigma$ for Algorithm 2 in Appendix A.4.2, and comparison with model-free LQR in Appendix A.4.3. We also compare the tracking performance of our proposed model-free PG4PI algorithm with model-free LQR, on the Chemical Reactor environment, in Figure 2. The code for this implementation is publicly available at `https://github.com/sharma1256/RL-optimal-pid`.

**General discussion.** As illustrated in Fig. 1, for Algorithms 1 and 2, the rewards accumulated during training rapidly converge to zero on both environments, indicating that both algorithm rapidly learn to stabilize the system and track the reference output. PPO, on the other hand, requires a large number of samples before it begins to learn to stabilize the system, and its tracking performance remains far worse than that of our proposed methods. The superior performance of Algorithms 1 and 2 is due to the fact that they directly search over the space of PID parameters and are guaranteed to converge to global optimality, while PPO searches over the much larger space of all policy parameterizations mapping states to PID control parameters and may only achieve a local optimum or saddle point of this much larger, highly nonconvex problem where gradient dominance likely does not hold. Finally, our ablation experiments studying the impact of $\sigma$ on performance of Algorithm 2 indicate that the choice of $\sigma$ does not have a noticeable impact on convergence in practice.

**Benchmarks vs. LQR.** We implement PG-based LQR algorithms from [18] on both the environments. From Figures 2 and 6 in the Appendix, we see that the PID control policy achieves zero steady-state tracking error in contrast to non-zero tracking error with the LQR policy, due to the inclusion of the integral parameter in the PID control policy. Importantly, LQR/LQG variants are known to be fragile to model error [15]. In contrast, PID policies can yield higher robustness margins in practice due to the output-feedback structure and low dimensionality. We demonstrate this in Fig. 7 in the Appendix, where a very small perturbation of the matrix $A$ ($\|\Delta A\| = 0.5$) breaks stability even in model-based LQR, while our model-free PI policy maintains stability and tracking.

# 6 Conclusion

In this paper, we proposed and studied a new optimization framework bridging the gap between PG methods for control and the PID control architecture. Building on this framework, we developed model-based and model-free PG methods for learning PID and PI control policies, respectively, then established convergence, global optimality, and sample complexity guarantees. Important future directions include extending our mathematical framework to encompass multi-objective trade-offs arising from classical control specifications or through mixed PID $\mathcal{H}_2/\mathcal{H}_\infty$ controller parameterizations, as well as incorporating hard constraints on system states or outputs, which will involve extending our theoretical framework to constrained optimization settings via safety-constrained exploration [42] or safety filters [12] to obtain safe PID policies.

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

# A  Appendix

Here, we present proofs of all the results in the paper, and additional details for each section.

## A.1  Proofs and Additional Details for Section 2

**Proof of Theorem 2.1.** From $y_t = Cx_t$, we can write

$$y_t^T Y y_t = x_t^T C^T Y C x_t = x_t^T C^T Y C x_t = g_t^T Q g_t. \tag{33}$$

Now, the PID control policy from (2) can be expressed as

$$u_t = -K_P C x_t - K_D C \frac{x_{k+1} - x_t}{\tau} - K_I z_t. \tag{34}$$

Substituting $x_{t+1} = Ax_t + Bu_t$, (5) can be further written as

$$u_t = -K_P C x_t - K_D C \frac{Ax_t + Bu_t - x_t}{\tau} - K_I z_t \tag{35}$$

Rearranging the above equation, we get

$$u_t = -\left[1 + \frac{K_D C B}{\tau}\right]^{-1}\left(\left(K_P C + K_D \frac{C(A-I)}{\tau}\right)x_t + K_I z_t\right). \tag{36}$$

Substituting the control policy $u_t$ from (36) into the dynamics $x_{t+1} = Ax_t + Bu_t$, we can write

$$x_{t+1} = Ax_t - B\left[1 + \frac{K_D C B}{\tau}\right]^{-1}\left(\left(K_P C + K_D \frac{C(A-I)}{\tau}\right)x_t + K_I z_t\right) \tag{37}$$

From $z_t = \sum_{i=0}^{t-1} y_i$, we can write $z_{t+1} = y_t + z_t = Cx_t + z_t$. Now, we have

$$
\begin{aligned}
g_{t+1} = \begin{bmatrix} x_{t+1} \\ z_{t+1} \end{bmatrix} &= \begin{bmatrix} Ax_t - B\left[1 + \frac{K_D C B}{\tau}\right]^{-1}\left(\left(K_P C + K_D \frac{C(A-I)}{\tau}\right)x_t + K_I z_t\right) \\ Cx_t + z_t \end{bmatrix} \\
&= \begin{bmatrix} Ax_t \\ Cx_t + z_t \end{bmatrix} + \begin{bmatrix} -B\left[1 + \frac{K_D C B}{\tau}\right]^{-1}\left(\left(K_P C + K_D \frac{C(A-I)}{\tau}\right)x_t + K_I z_t\right) \\ 0 \end{bmatrix}, \\
&= \begin{bmatrix} Ax_t \\ Cx_t + z_t \end{bmatrix} - \begin{bmatrix} B \\ 0 \end{bmatrix}\left[1 + \frac{K_D C B}{\tau}\right]^{-1}\left(\left(K_P C + K_D \frac{C(A-I)}{\tau}\right)x_t + K_I z_t\right).
\end{aligned} \tag{38}
$$

Recalling from (6) that

$$K = \left[1 + \frac{K_D C B}{\tau}\right]^{-1}\left[K_P C + K_D \frac{C(A-I)}{\tau} \quad K_I\right], \tag{39}$$

we can write

$$g_{t+1} = \bar{A}g_t + \bar{B}u_t, \tag{40}$$

where $\bar{A} = \begin{bmatrix} A & 0 \\ C & I \end{bmatrix}, \bar{B} = \begin{bmatrix} B \\ 0 \end{bmatrix}$, and $u_t = -Kg_t$. Finally, from (33), we have that

$$J(K) = \mathbb{E}\left[\sum_{t=0}^{\infty} y_t^T Y y_t + u_t^T R u_t\right] = \mathbb{E}\left[\sum_{t=0}^{\infty} g_t^T Q g_t + u_t^T R u_t\right], \tag{41}$$

whence the PID control problem can be rewritten as

$$
\begin{aligned}
\text{minimize} \quad & J(\tilde{K}) = \mathbb{E}\left[\sum_{t=0}^{\infty} g_t^T Q g_t + u_t^T R u_t\right] \\
\text{subject to} \quad & g_{t+1} = \bar{A}x_t + \bar{B}u_t, \quad g_0 \sim \mathcal{D},
\end{aligned} \tag{42}
$$

with $u_t = -Kg_t$ where $K$ is obtained from PID parameters $\tilde{K}$ using (6). □

## A.2 Proofs and Additional Details for Section 3

In this section, we provide a detailed proofs of each of the results of Section 3. We begin by defining the cost-to-go at a fixed initial state $g_0$ by

$$V_K(g_0) = g_0^T P_K g_0. \tag{43}$$

From (10), the cost-to-go can also be written as

$$V_K(g_0) = g_0^T (A_K^T P_K A_K + Q + K^T RK) g_0. \tag{44}$$

Notice that $J(K) = \mathbb{E}_{g_0 \sim \mathcal{D}} [V_K(g_0)]$. This cost-to-go expression will be useful when calculating gradients with respect to the the individual PID control parameters $K_P, K_I, K_D$ in the following gradient expressions.

### A.2.1 Proof in subsection 3.1

En route to proving Theorem 3.1, we establish the following helper lemmas that provide derivatives with respect to the parameters in $\tilde{K}$ of certain expressions involving $A_K$ and $K^T RK$ that arise in the main results to follow. We first begin with the derivative of the expression $v^T A_K g_0$, where $v$ is an arbitrarily chosen vector with appropriate dimensions and $g_0$ is the initial state.

**Lemma A.1** (Helper Lemma for Theorem 3.1). *Let $v$ be a vector of suitable dimensions such that $v^T A_K g_0$ is a scalar, where $g_0$ is the initial condition. Then, the matrix derivatives $\nabla_{K_i} v^T A_K g_0$, for all $i \in \{P, I, D\}$ are given as*

$$\nabla_{K_P} v^T A_K g_0 = - F_D \bar{B}^T v g_0^T T_x^T C^T, \tag{45}$$

$$\nabla_{K_I} v^T A_K g_0 = - F_D \bar{B}^T v g_0^T T_z^T, \tag{46}$$

$$\nabla_{K_D} v^T A_K g_0 = \alpha_1 \big[ \bar{B}^T v g_0^T (\alpha_2 \alpha_3 - T_x^T (A - I)^T C^T) \big], \tag{47}$$

*where, $T_x = [I_n \quad 0_{n\times 1}], T_z = [0_{1\times n} \quad 1], \alpha_1 = (\tau + K_D CB)^{-1}, \alpha_2 = (\tau + K_D CB)^{-1}(\tau T_x^T C^T K_P + T_x^T (A - I)^T C^T K_D + \tau T_z^T K_D), \alpha_3 = CB,$ and $F_D = [1 + \frac{K_D CB}{\tau}]^{-1}.$*

*Proof.* From (8), we know that $g_1 = A_K g_0$, and therefore

$$x_1 = (A - BF_D K_P C - BF_D K_D \frac{C(A - I)}{\tau}) x_0 - BF_D K_I z_0, \tag{48}$$

$$z_1 = Cx_0 + z_0.$$

We note that the variables $F_D, K_P, K_I, K_D$ are scalars, and, that the matrix $P_K$ is symmetric. Since we know that $z_1$ does not contain any of the $K_P, K_I, K_D$ parameters, we can write

$$\nabla_{K_i} v^T A_K g_0 = \nabla_{K_i} v_{1:n}^T x_1, \text{ for all } i \in \{P, I, D\}, \tag{49}$$

where $v_{1:n}$ denotes the the first $n$-elements of $v$. Then, we can write

$$\nabla_{K_P} v_{1:n}^T x_1 = -\nabla_{K_P} (v_{1:n}^T BF_D K_P Cx_0),$$

where

$$\nabla_{K_P} (v_{1:n}^T BF_D K_P Cx_0) = v_{1:n}^T BF_D Cx_0,$$

due to the fact that $K_P$ is a scalar. Observe that, $v_{1:n}^T B = v^T \bar{B}$. Thus,

$$\nabla_{K_P} v^T A_K g_0 = -v^T \bar{B} F_D Cx_0. \tag{50}$$

Since the above expression is a scalar, we transpose the right hand side to obtain

$$\nabla_{K_P} v^T A_K g_0 = -F_D \bar{B}^T v x_0^T C^T. \tag{51}$$

From the fact that $x_0 = T_x g_0$, we can write

$$\nabla_{K_P} v^T A_K g_0 = -F_D \bar{B}^T v g_0^T T_x^T C^T. \tag{52}$$

Similarly, from (49), we can write

$$\nabla_{K_I} v_{1:n}^T x_1 = -\nabla_{K_I} (v_{1:n}^T BF_D K_I z_0). \tag{53}$$

Then, since $z_0 = T_z g_0$, we can obtain the gradient
$$\nabla_{K_I} v^T A_K g_0 = -F_D \bar{B}^T v g_0^T T_z^T. \tag{54}$$
We can further expand the above as

$$\nabla_{K_D} v_{1:n}^T x_1 = -\nabla_{K_D} (v_{1:n}^T B \frac{1}{1 + \frac{K_D CB}{\tau}} (K_P C x_0 K_D \frac{C(A-I)}{\tau} x_0 - K_I z_0))$$

$$= -[v_{1:n}^T B (K_P C x_0 K_D \frac{C(A-I)}{\tau} x_0 + K_I z_0)] \nabla_{K_D} \frac{1}{1 + \frac{K_D CB}{\tau}}$$

$$\quad - [v_{1:n}^T B \frac{1}{1 + \frac{K_D CB}{\tau}} (\nabla_{K_D} (K_P C x_0 K_D \frac{C(A-I)}{\tau} x_0 - K_I z_0))],$$

$$= (v^T \bar{B} (K_P C x_0 K_D \frac{C(A-I)}{\tau} x_0 - K_I z_0)) \frac{CB}{\tau} \frac{1}{(1 + \frac{K_D CB}{\tau})^2}$$

$$\quad - (v^T \bar{B} \frac{1}{1 + \frac{K_D CB}{\tau}} (\frac{C(A-I)}{\tau} x_0)).$$

Simplifying the above expression, we obtain
$$\nabla_{K_D} v^T A_K g_0 = \alpha_1 [\bar{B}^T v g_0^T (\alpha_2 \alpha_3 - T_x^T (A-I)^T C^T)]. \tag{55}$$
$\square$

We next provide derivatives of the expression $g_0^T K^T R K g_0$ with respect to $\tilde{K}$, where $g_0$ is the initial state.

**Lemma A.2** (Helper Lemma for Theorem 3.1). *The matrix derivatives* $\nabla_{K_i} g_0^T K^T R K g_0$, *for each* $i \in \{P, I, D\}$ *are*

$$\nabla_{K_P} g_0^T K^T R K g_0 = 2 F_D R K g_0 g_0^T T_x^T C^T, \tag{56}$$

$$\nabla_{K_I} g_0^T K^T R K g_0 = 2 F_D R K g_0 g_0^T T_z^T, \tag{57}$$

$$\nabla_{K_D} g_0^T K^T R K g_0 = 2\alpha_1 [R K g_0 g_0^T [T_x^T (A-I)^T C^T - \alpha_2 \alpha_3)]],$$

*where,* $T_x = [I_n \quad 0_{n \times 1}], T_z = [0_{1 \times n} \quad 1], \alpha_1 = (\tau + K_D CB)^{-1}, \alpha_2 = (\tau + K_D CB)^{-1} (\tau T_x^T C^T K_P + T_x^T (A-I)^T C^T K_D + \tau T_z^T K_D), \alpha_3 = CB,$ *and* $F_D = [1 + \frac{K_D CB}{\tau}]^{-1}.$

*Proof.* We begin by noting that $F_D, K_P, K_I, K_D, R$ are scalars. Now, for all $i \in \{P, I, D\}$, we can write
$$\nabla_{K_i} g_0^T K^T R K g_0 = 2 \nabla_{K_i} w^T K g_0, \tag{58}$$
where $w$ is given by $w = R^T K g_0 = R K g_0$. Then, from $K g_0 = F_D (K_P C + K_D \frac{C(A-I)}{\tau}) x_0 - F_D K_I z_0$, we get
$$\nabla_{K_P} w^T K g_0 = \nabla_{K_P} w^T F_D K_P C x_0 = w^T F_D C x_0 = g_0^T K^T R F_D C x_0. \tag{59}$$
Since the above gradient is a scalar, we take transpose on the right hand side and get
$$\nabla_{K_P} w^T K g_0 = F_D R K g_0 x_0^T C^T. \tag{60}$$
We note that $x_0 = T_x g_0$, and, therefore we can write
$$\nabla_{K_P} w^T K g_0 = F_D R K g_0 g_0^T T_x^T C^T. \tag{61}$$

Similarly, we can write
$$\nabla_{K_I} w^T K g_0 = F_D R K g_0 g_0^T T_z^T. \tag{62}$$

We now calculate $\nabla_{K_D} w^T K g_0$. We first expand the gradient as follows:

$$\nabla_{K_D} w^T K g_0 = w^T ((K_P C + K_D \frac{C(A-I)}{\tau}) x_0 - F_D K_I z_0) \nabla_{K_D} F_D$$

$$\quad + F_D \nabla_{K_D} w^T ((K_P C + K_D \frac{C(A-I)}{\tau}) x_0 - K_I z_0)$$

$$= -w^T ((K_P C + K_D \frac{C(A-I)}{\tau}) x_0 - F_D K_I z_0) \frac{\tau CB}{(\tau + CB)^2}$$

$$\quad + w^T F_D (\frac{C(A-I)}{\tau}) x_0.$$

Expanding these terms, we finally get the following expression

$$\nabla_{K_D} w^T K g_0 = \alpha_1 \big[ R^T K g_0 g_0^T [T_x^T (A - I)^T C^T - \alpha_2 \alpha_3)] \big],$$

where the expressions for $\alpha_1, \alpha_2, \alpha_3$ are as defined in statement of Theorem 3.1. $\qquad\square$

We are now ready to prove the Theorem 3.1.

***Proof of Theorem 3.1.*** For each $i \in \{P, I, D\}$, we have

$$
\begin{aligned}
\nabla_{K_i} V_K(g_0) &= \nabla_{K_i} g_0^T P_K g_0 \\
&= \nabla_{K_i} g_0^T A_K^T P_K A_K g_0 + \nabla_{K_i} g_0^T Q g_0 + \nabla_{K_i} g_0^T K^T R K g_0.
\end{aligned}
\tag{63}
$$

In the above expression, we first focus on $\nabla_{K_i} g_0^T A_K^T P_K A_K g_0$, which we can rewrite as

$$\nabla_{K_i} g_0^T A_K^T P_K A_K g_0 = 2 \nabla_{K_i} v^T A_K g_0 + \nabla_{K_i} g_1^T P_K g_1, \tag{64}$$

where $v = P_K^T A_K g_0$ and $g_1 = A_K g_0$.

However, we note that $V_K(g_1) = g_1^T P_K g_1$ and thus

$$\nabla_{K_i} g_0^T A_K^T P_K A_K g_0 = 2 \nabla_{K_i} v^T A_K g_0 + \nabla_{K_i} V_K(g_1). \tag{65}$$

Then, from Lemma A.1, we obtain

$$
\begin{aligned}
\nabla_{K_P} v^T A_K g_0 &= -F_D \bar{B}^T P_K A_K g_0 g_0^T T_x^T C^T, \\
\nabla_{K_I} v^T A_K g_0 &= -F_D \bar{B}^T P_K A_K g_0 g_0^T T_z^T, \\
\nabla_{K_D} v^T A_K g_0 &= \alpha_1 \big[ \bar{B}^T P_K A_K g_0 g_0^T (\alpha_2 \alpha_3 - T_x^T (A - I)^T C^T) \big]
\end{aligned}
$$

Moreover, we know from Lemma A.2 that

$$
\begin{aligned}
\nabla_{K_P} g_0^T K^T R K g_0 &= 2 F_D R K g_0 g_0^T T_x^T C^T, \\
\nabla_{K_I} g_0^T K^T R K g_0 &= 2 F_D R K g_0 g_0^T T_z^T, \\
\nabla_{K_D} g_0^T K^T R K g_0 &= 2 \alpha_1 \big[ R K g_0 g_0^T [T_x^T (A - I)^T C^T - \alpha_2 \alpha_3)] \big].
\end{aligned}
$$

Combining these expressions, we can write

$$\nabla_{K_P} V_K(g_0) = -2 F_D \bar{B}^T P_K A_K g_0 g_0^T T_x^T C^T + \nabla_{K_P} V_K(g_1) + 2 F_D R K g_0 g_0^T T_x^T C^T. \tag{66}$$

Note that $\nabla_{K_P} V_K(g_1)$ can again be written as

$$\nabla_{K_P} V_K(g_1) = -2 F_D \bar{B}^T P_K A_K g_1 g_1^T T_x^T C^T + \nabla_{K_P} V_K(g_2) + 2 F_D R K g_1 g_1^T T_x^T C^T. \tag{67}$$

Proceeding in this manner, we have

$$\nabla_{K_P} V_K(g_0) = \sum_{i=0}^{\infty} \{ -2 F_D \bar{B}^T P_K A_K g_i g_i^T T_x^T C^T + 2 F_D R K g_i g_i^T T_x^T C^T \}. \tag{68}$$

In light of this, we can write

$$\nabla_{K_P} J(K) = 2 (-F_D \bar{B}^T P_K A_K + F_D R K) \Sigma_K T_x^T C^T. \tag{69}$$

Noting that

$$E_K = (R + \bar{B}^T P_K \bar{B}) K - \bar{B}^T P_K \bar{A} = -\bar{B}^T P_K A_K + R K, \tag{70}$$

we obtain

$$\nabla_{K_P} J(K) = 2 F_D E_K \Sigma_K T_x^T C^T. \tag{71}$$

Similarly, we can obtain

$$
\begin{aligned}
\nabla_{K_I} J(K) &= 2 F_D E_K \Sigma_K T_z^T, \\
\nabla_{K_D} J(K) &= 2 F_D E_K \Sigma_K (\alpha_1 T_x^T (A - I)^T C^T - \alpha_2 \alpha_3).
\end{aligned}
$$

$\qquad\square$

### A.2.2   Estimation of Matrices $(A, B, P_K)$ in Algorithm 1

In the next two subsections, we provide details regarding the estimation of matrices $(A, B, P_K)$. We begin by describing a standard system identification approach that we use to obtain matrices $(A, B)$, which are required to estimate the gradients utilized in the update rules of Algorithm 1.

**System Identification for** $(A, B)$. We follow the standard least squares system identification approach in [27] to estimate the matrices $A$ and $B$. Given trajectory data $x_0, u_0, x_1, u_1, \ldots, x_{t+1}, u_{t+1}$ collected from the system, for some $t > 0$, and taking $k \le t$, we first consider the following dynamical equation

$$X_{k+1} = AX_k + BU_k, \tag{72}$$

where

$$X_k = [x_0 \ x_1 \ \ldots \ x_{k-1}], \tag{73}$$
$$U_k = [u_0 \ u_1 \ \ldots \ u_{k-1}]. \tag{74}$$

We then define the least squares estimation error as

$$E(A, B) = \|X_{k+1} - (AX_k + BU_k)\|^2. \tag{75}$$

Defining

$$\theta := [A \ B], M := \begin{bmatrix} X_k \\ U_k \end{bmatrix}, \tag{76}$$

and noting that $X_{k+1} = \theta M$, we obtain the following estimate

$$\theta = X_{k+1}M^T(MM^T)^{-1} \tag{77}$$

for the system matrices $(A, B)$. With the estimation technique for $(A, B)$ described, we estimate the matrix $P_{K_t}$ by the method described in the next section to ultimately estimate the gradients $\nabla_{K_i} J(\tilde{K})$, for each $i \in \{P, I, D\}$.

**Iterative Method for Estimating** $P_K$. Consider the following Riccati equation for our reformulated system in Proposition 2.1, where $K$ is obtained from $\tilde{K}$ using (6):

$$P_K = (\bar{A} - \bar{B}K)^T P_K(\bar{A} - \bar{B}K) + Q + K^T RK. \tag{78}$$

Right multiplying the above equation by $g_t^T$, and left multiplying it by $g_t$, we can write

$$g_t^T P_K g_t = g_t^T (\bar{A} - \bar{B}K)^T P_K(\bar{A} - \bar{B}K)g_t + g_t^T Q g_t + g_t^T K^T RK g_t. \tag{79}$$

Using $g_{t+1} = \bar{A}g_t + \bar{B}u_t$, $u_t = -Kg_t$, we rearrange the above expression and write

$$g_t^T P_K g_t - g_{t+1}^T P_K g_{t+1} = g_t^T Q g_t + g_t^T K^T RK g_t. \tag{80}$$

We can now use (80) to estimate the matrix $P_K$ by collecting sufficient samples of $g_t$. Consider

$$q_t := g_t^T Q g_t + g_t^T K^T RK g_t, \tag{81}$$

and the vector $P_K^v \in \mathbb{R}^{(n+1)^2}$ is obtained by vertically concatenating the column vectors of $P_K$. Before defining the data set for the estimation $P_K$, we review some preliminary results from matrix theory.

For a vector $w \in \mathbb{R}^n$ and a matrix $U \in \mathbb{R}^{n \times n}$, we know that $w^T U w = \sum_{i=0}^{n+1} \sum_{j=0}^{n+1} U_{i,j} w_i w_j$, where $U_{i,j}$ is the $(i, j)$-th element of $U$, and $w_i$ is the $i$-th element of the vector $w$. Consequently, we can write $w^T U w = \sum_{i=0}^{n^2} U_i^v w_i^v$, where $U_i^v$ and $w_i^v$ are the $i$-th elements of the vectorized version of matrix $U$ and the Kronecker product $w \otimes w$, respectively.

Using the above facts, we can write

$$g_t^T P_K g_t - g_{t+1}^T P_K g_{t+1} = P_K^{v^T}(g_t \otimes g_t - g_{t+1} \otimes g_{t+1}), \tag{82}$$

from which we obtain

$$(g_t \otimes g_t - g_{t+1} \otimes g_{t+1})^T P_K^v = q_t. \tag{83}$$

We now define matrices $\mathbb{G}_k$ and $\mathbb{O}_k$, for $k \geq \frac{n(n+1)}{2} + n + 1$, as

$$\mathbb{G}_k = [g_0 \otimes g_t \; g_1 \otimes g_1 \; \cdots \; g_{k-1} \otimes g_{k-1}], \qquad (84)$$

$$\mathbb{O}_k = [q_0 \; q_1 \; \cdots \; q_{k-1}]. \qquad (85)$$

Observe that these matrices can be constructed from the set $X_k$, defined in (73). Then, we can write

$$\hat{\mathbb{G}}_k v = y_k, \qquad (86)$$

where $v = P_{K_t}^{v^T}$, $\hat{\mathbb{G}}_k := (\mathbb{G}_k - \mathbb{G}_{k+1})^T$, and $y_k = \mathbb{O}_k$. Finally, we minimize the squared error

$$E(v) = \|y_k - \hat{\mathbb{G}}_k v\|^2, \qquad (87)$$

to obtain

$$v = (\hat{\mathbb{G}}_k^T \hat{\mathbb{G}}_k)^{-1} \hat{\mathbb{G}}_k^T y_k. \qquad (88)$$

In this manner, we estimate the vectorized version of $P_K$.

### A.2.3 Proofs in Subsection 3.2

In this subsection, we provide proofs of results in Subsection 3.2 on calculating policy gradients in the model-free setting that are used in the update rules of Algorithm 2. We first begin with the proof of Theorem 3.2, where expressions for the score functions with respect to each of the $PID$ parameters are provided.

***Proof of Theorem 3.2.*** Given $\mu_K(g) = -Kg = -[1 + \frac{K_D C B}{\tau}]^{-1}[(K_P C + \frac{K_D C(A-I)}{\tau})x - K_I z]$, it immediately follows that

$$\nabla_{K_P} \mu_K(g) = -\tau(\tau + K_D C B)^{-1} C T_x g \qquad (89)$$

$$\nabla_{K_I} \mu_K(g) = -\tau(\tau + K_D C B)^{-1} T_z g. \qquad (90)$$

To compute $\nabla_{K_D} \mu_K(g)$, we first write

$$(1 + \frac{K_D C B}{\tau}) \mu_K(g) = -(K_P C + \frac{K_D C(A-I)}{\tau})x + K_I z.$$

Then, applying the operator $\nabla_{K_D}$ on both the sides, we obtain

$$\nabla_{K_D}\{(1 + \frac{K_D C B}{\tau}) \mu_K(g)\} = -\frac{C(A-I)x}{\tau},$$

or, equivalently,

$$\frac{\mu_K(g) C B}{\tau} + (1 + \frac{K_D C B}{\tau}) \nabla_{K_D} \mu_K(g) = -\frac{C(A-I)x}{\tau}.$$

Rearranging the above expression, we get

$$\nabla_{K_D} \mu_K(g) = -(\tau + K_D C B)^{-1}(C(A-I)x + \mu_K(g) C B).$$

$\square$

We now move on to proving Corollary 3.3, where we establish expressions for the score function with respect to $PI$ parameters that do not require the knowledge of matrices $(A, B)$.

***Proof of Corollary 3.3.*** From (18), we know that

$$\nabla_{K_P} \log \pi_K(u|g) = \frac{u - \mu_K(g)}{\sigma^2} \nabla_{K_P} \mu_K(g), \qquad (91)$$

$$\nabla_{K_I} \log \pi_K(u|g) = \frac{u - \mu_K(g)}{\sigma^2} \nabla_{K_I} \mu_K(g). \qquad (92)$$

From Theorem 3.2 with $K_D = 0$, we get

$$\nabla_{K_P} \log \pi_K(u|g) = -\frac{u - \mu_K(g)}{\sigma^2} g^T T_x^T C^T, \qquad (93)$$

$$\nabla_{K_I} \log \pi_K(u|g) = -\frac{u - \mu_K(g)}{\sigma^2} g^T T_z^T. \qquad (94)$$

$\square$

## A.3 Proofs and Additional Details for Section 4

We begin with proving that the objective of problem 3 enjoys the gradient dominance property in $\tilde{K}$.

***Proof of Theorem 4.1.*** We first define the set of stabilizing control parameters as

$$\mathcal{K} := \left\{ K \in \mathbb{R}^{1 \times n+1} : \rho(\bar{A} - \bar{B}K) < 1 \right\}. \tag{95}$$

Then, we know from Lemma 3 in [18] that the objective in Proposition 2.1 with the control policy in (5), enjoys the gradient dominance property in $K$, that is,

$$|J(K) - J(K^*)| \le \alpha \|\nabla_K J(K)\|^2, \text{ for all } K \in \mathcal{K}, \tag{96}$$

where $K, K^*$ are obtained from $\tilde{K}, \tilde{K}^*$ using (6), and, $\alpha = \|\Sigma_{K^*}\| \left[\sigma_{min}(\Sigma_K)^2 R\right]^{-1}$.

Since, we can replace $J(K)$ with $J(\tilde{K})$ Further, we note that $\nabla_{\tilde{K}} J(K) = [\nabla_{K_P} J(K) \quad \nabla_{K_I} J(K) \quad \nabla_{K_D} J(K)]^T$. We can rewrite $\frac{dJ(K)}{dK}$, using the chain rule, as

$$\frac{dJ(K)}{dK} = (\nabla_K K_P)\nabla_{K_P} J(\tilde{K}) + (\nabla_K K_I)\nabla_{K_I} J(\tilde{K}) + (\nabla_K K_D)\nabla_{K_D} J(\tilde{K}),$$

or, equivalently,

$$\nabla_K J(K) = m_P \nabla_{K_P} J(\tilde{K}) + m_I \nabla_{K_I} J(\tilde{K}) + m_D \nabla_{K_D} J(\tilde{K}),$$

where the matrices $m_P, m_I, m_D$ are given by $m_P = \nabla_K K_P$, $m_I = \nabla_K K_I$ and $m_D = \nabla_K K_D$. Then, we can write the following:

$$\nabla_K J(K) = \nabla_{\tilde{K}} J(\tilde{K})^T M_{PID}, \tag{97}$$

where

$$M_{PID} = \begin{bmatrix} m_P \\ m_I \\ m_D \end{bmatrix}. \tag{98}$$

Applying the norm operator on both sides, we obtain $\|\nabla_K J(K)\| = \left\| \nabla_{\tilde{K}} J(\tilde{K})^T M_{PID} \right\|$. Then, using the face that $\|\nabla_K J(K)\|^2 \le \|M_{PID}\|^2 \left\| \nabla_{\tilde{K}} J(\tilde{K}) \right\|^2$, we have

$$|J(K) - J(K^*)| \le \alpha \|M_{PID}\|^2 \|\nabla_{\tilde{K}} J(\tilde{K})\|^2.$$

Finally, from the equivalence $J(\tilde{K}) = J(K)$, we have

$$\left| J(\tilde{K}) - J(\tilde{K}^*) \right| \le \alpha \|M_{PID}\|^2 \|\nabla_{\tilde{K}} J(\tilde{K})\|^2.$$

$\square$

### A.3.1 Proofs in subsection 4.1

In this section, we present the proof of Theorem 4.2. First, we define relevant variables from Theorem 3 in [13] for our formulation and then adapt their sample complexity result for our setting. We begin with the following conditions on the step size $\eta$. Let $\eta > 0$ be such that

$$\begin{aligned} &\rho(\bar{A} - \bar{B}K_\eta) < 1, \\ &Tr(E_K^T \nabla_K J_\mu^D(K)\Sigma_{K_\eta}(1 - \eta a \Sigma_K)) > 0, . \end{aligned} \tag{99}$$

where $K_\eta = K - \eta \nabla_K J_\mu^D(K)$ and $a = \lambda_{max}(R + \bar{B}^T P_K \bar{B})$.

***Proof of Theorem 4.2.*** We note that $\nabla_K J_\mu^D(K) = 2E_K \Sigma_K$, where $\Sigma_K = \mathbb{E} \sum_{i=0}^\infty g_i g_i^T$ and $E_K$ is given (14). Comparing to the gradient of the function $f(K)$ from [13, Proposition 1], we have $Y := \Sigma_K, Y_\eta := \Sigma_{K_\eta}, A_{K_\eta} = \bar{A} - \bar{B}K_\eta$ and $\phi_\eta = \frac{1}{2}Tr(E_K^T \nabla_K J_\mu^D(K)\Sigma_{K_\eta}(1 - \eta a \Sigma_K))$ in our setting. Then, from [13, Theorem 3] , for all $t \ge 0$,

$$J_\mu^D(K_t) - J_\mu^D(K^*) \le \kappa^t \left( J_\mu^D(K_0) - J_\mu^D(K^*) \right).$$

We note that $K$ is a function of $\tilde{K}$ by (6), and, $\tilde{K}^* := [K_P^* \quad K_I^* \quad K_D^*]^T$ corresponds to $K^*$, the global minimizer of $J_\mu^D(K)$. Thus, with a slight abuse in notation, we can write

$$J_\mu^D(\tilde{K}_t) - J_\mu^D(\tilde{K}^*) \leq \kappa^t \left( J_\mu^D(\tilde{K}_0) - J_\mu^D(\tilde{K}^*) \right).$$

$\square$

We now move on to convergence results for model-free case.

### A.3.2 Proofs in Subsection 4.2

First, recall that for the model-free PI algorithm where $K_D = 0$, we modify the $PI$ control parameter to be

$$\tilde{K} := \begin{bmatrix} K_P \\ K_I \end{bmatrix}. \tag{100}$$

Consequently, from (6) with $K_D = 0$, we get

$$K = [K_P C \quad K_I], \tag{101}$$

or equivalently,

$$K = S\tilde{K}, \tag{102}$$

where

$$S = [C \quad 1]. \tag{103}$$

Further, we define the matrix $M_{PI}$ as

$$M_{PI} := \begin{bmatrix} \nabla_K K_P \\ \nabla_K K_I \end{bmatrix}. \tag{104}$$

Second, under Assumption 4.3, the stochastic policy in (16) becomes

$$\pi_K(u|g_t) = \frac{1}{\left(\frac{\sigma}{t+1}\right)\sqrt{2\pi}} e^{-\frac{(u - \mu_{\tilde{K}}(g_t))^T (u - \mu_{\tilde{K}}(g_t))}{2\left(\frac{\sigma}{t+1}\right)^2}}, \tag{105}$$

that is, $u_t \sim \mathcal{N}(\mu_{\tilde{K}}(g_t), \sigma^2/(t+1)^2)$, for all $t \geq 0$. We can also write the following:

$$u_t = \mu_{\tilde{K}}(g_t) + \eta_t, \tag{106}$$

where $\eta_t \sim \mathcal{N}(0, \sigma^2/(t+1)^2)$, for all $t \geq 0$. We note that the $\{\eta_i\}_{i=0}^\infty$ are i.i.d. random variables and independent of $g_0$.

We now present helper results that we will use in the proof of Theorem 4.7. We begin by obtaining a uniform bound over the magnitude $\left| J_\pi^G(\tilde{K}) - J_\mu^D(\tilde{K}) \right|$.

**Theorem A.3.** *The costs $J_K$ and $J_K^G$ satisfy*

$$\left| J_\mu^D(\tilde{K}) - J_\pi^G(\tilde{K}) \right| \leq \frac{\|R\|\pi^2\sigma^2}{6}, \text{ for all } \tilde{K} \in \tilde{\mathcal{K}}. \tag{107}$$

*Proof.* Since the control input $u_t$ depends on $g_0$ through $g_t$ and $\{\eta_i\}_{i=0}^\infty$, the expectation in (25) will be with respect to $\{\eta_i\}_{i=0}^\infty$ and $g_0$. Furthermore, by (6) we can write $J_\pi^G(K) = J_\pi^G(\tilde{K})$ and $J_\mu^D(K) = J_\mu^D(\tilde{K})$. We expand the expression for $J_\pi^G(K)$ to get

$$J_\pi^G(K) = \mathbb{E}_{\{\eta_i\}_{i=0}^\infty} \mathbb{E}_{g_0} \left[ \sum_{j=0}^\infty y_j^T Y y_j + (\mu_K(g_j) + \eta_j)^T R(\mu_K(g_j) + \eta_j) \right]. \tag{108}$$

Further expanding this expression gives

$$J_\pi^G(K) = \mathbb{E}_{\{\eta_i\}_{i=0}^\infty} \mathbb{E}_{g_0} \left[ \sum_{j=0}^\infty y_j^T Y y_j + \mu_K(g_j)^T R\mu_K(g_j) + 2\eta_j^T R\mu_K(g_j) + \eta_j^T R\eta_j \right]. \tag{109}$$

We observe that the expectation in $J_\mu^D(\tilde{K})$ is with respect to $\mu_{\tilde{K}} = -Kg$, with $K$ given by (6), which implies that the expectation is taken with respect to $g_0 \sim \mathcal{D}$, and thus

$$J_\mu^D(\tilde{K}) = \mathbb{E}_{g_0}[\sum_{j=0}^{\infty} y_j^T Y y_j + \mu_{\tilde{K}}(g_j)^T R \mu_{\tilde{K}}(g_j)]. \tag{110}$$

Thus, we can further write the following:

$$J_\pi^G(K) = J_\mu^D(K) + \mathbb{E}_{\{\eta_i\}_{i=0}^{\infty}} \mathbb{E}_{g_0}[\sum_{j=0}^{\infty} 2\eta_j^T R \mu_{\tilde{K}}(g_j) + \eta_j^T R \eta_j]. \tag{111}$$

First, since $\{\eta_i\}$s are independent of $g_0$, we have

$$\mathbb{E}_{\{\eta_i\}_{i=0}^{\infty}} \mathbb{E}_{g_0}[\sum_{j=0}^{\infty} \eta_j^T R \eta_j] = \mathbb{E}_{\{\eta_i\}_{i=0}^{\infty}}[\sum_{j=0}^{\infty} \eta_j^T R \eta_j] \tag{112}$$

Then, from definition of variance and the fact the $\mathbb{E}_{\eta_j}[\eta_j] = 0$, for all $j \geq 0$, we get

$$\lim_{T \to \infty} \mathbb{E}_{\{\eta_i\}_{i=0}^{\infty}}[\sum_{j=0}^{T} \eta_j^T R \eta_j] = R \lim_{T \to \infty} \sum_{t=0}^{T} \frac{\sigma^2}{(t+1)^2}. \tag{113}$$

Since $\lim_{T \to \infty} \sum_{t=0}^{T} \frac{1}{(t+1)^2} = \frac{\pi^2}{6}$, we therefore have

$$\lim_{T \to \infty} \mathbb{E}_{\{\eta_i\}_{i=0}^{\infty}}[\sum_{j=0}^{T} \eta_j^T R \eta_j] = \frac{R\sigma^2\pi^2}{6}. \tag{114}$$

Second, we examine the expression $\lim_{T \to \infty} \mathbb{E}_{\{\eta_i\}_{i=0}^{\infty}} \mathbb{E}_{g_0}[\sum_{j=0}^{T} 2\eta_j^T R \mu_{\tilde{K}}(g_j)]$ and focus on the $j$-th term, that is, $\mathbb{E}_{\{\eta_i\}_{i=0}^{\infty}} \mathbb{E}_{g_0}[2\eta_j^T R \mu_{\tilde{K}}(g_j)]$.

We note that $\{\eta_i\}$ are i.i.d. and that the control input $\mu_{\tilde{K}}(g_j)$ only depends on the first $j-1$ random variables. We therefore have

$$\mathbb{E}_{\{\eta_i\}_{i=0}^{\infty}} \mathbb{E}_{g_0}[2\eta_j^T R \mu_{\tilde{K}}(g_j)] = 2\mathbb{E}_{\eta_j}[\eta_j^T] \mathbb{E}_{\{\eta_i\}_{i=0}^{j-1}} \mathbb{E}_{g_0}[R\mu_{\tilde{K}}(g_j)]. \tag{115}$$

However, since $\mathbb{E}_{\eta_j}[\eta_j^T] = 0$, we conclude that $\mathbb{E}_{\{\eta_i\}_{i=0}^{\infty}} \mathbb{E}_{g_0}[2\eta_j^T R \mu_{\tilde{K}}(g_j)] = 0$ and thus

$$\lim_{T \to \infty} \mathbb{E}_{\{\eta_i\}_{i=0}^{\infty}} \mathbb{E}_{g_0}[\sum_{j=0}^{T} 2\eta_j^T R \mu_{\tilde{K}}(g_j)] = 0. \tag{116}$$

Therefore we can rewrite (111) using (113) and (116) as

$$J_\pi^G(\tilde{K}) - J_\mu^D(\tilde{K}) = J_\pi^G(K) - J_\mu^D(K) = \frac{R\sigma^2\pi^2}{6}. \tag{117}$$

Taking norms on both sides in (117) and applying the triangle inequality finally establishes that

$$\left| J_\pi^G(\tilde{K}) - J_\mu^D(\tilde{K}) \right| \leq \frac{\|R\|\sigma^2\pi^2}{6}, \text{ for all } \tilde{K} \in \tilde{\mathcal{K}}. \tag{118}$$

$\square$

We now show that the added noise in the stochastic policy has no effect on the gradient.

**Theorem A.4.** *The gradients $\nabla_{\tilde{K}} J_\mu^D$ and $\nabla_{\tilde{K}} J_\pi^G$ are equivalent for all $\tilde{K} \in \tilde{\mathcal{K}}$, that is,*

$$\nabla_{\tilde{K}} J_\mu^D(\tilde{K}) = \nabla_{\tilde{K}} J_\pi^G(\tilde{K}). \tag{119}$$

*Proof.* From (117), we know that

$$J_\pi^G(\tilde{K}) - J_\mu^D(\tilde{K}) = \frac{R\sigma^2\pi^2}{6}, \tag{120}$$

for all $\tilde{K} \in \tilde{\mathcal{K}}$. Applying the operator $\nabla_{\tilde{K}}$ on both sides, we immediately obtain the desired equality as

$$\nabla_{\tilde{K}} J_\mu^D - \nabla_{\tilde{K}} J_\pi^G = 0, \text{ for all } \tilde{K} \in \tilde{\mathcal{K}}. \tag{121}$$

$\square$

Leveraging the last two results, we now bound the optimality gap arising due to use of the stochastic policy (16).

**Theorem A.5.** *Suppose* $\tilde{K}_G^* \in \arg\min_{\tilde{K}}\{J_\pi^G(\tilde{K})\}$ *and* $\tilde{K}^* \in \arg\min_{\tilde{K}} J_\mu^D(\tilde{K})$. *We have*

$$\left| J_\pi^G(\tilde{K}_G^*) - J_\mu^D(\tilde{K}^*) \right| \leq \frac{\|R\|\sigma^2\pi^2}{3}. \tag{122}$$

*Proof.* Consider the following:

$$\begin{aligned}
\left| J_\pi^G(\tilde{K}_G^*) - J_\mu^D(\tilde{K}^*) \right| &= \left| J_\pi^G(\tilde{K}_G^*) - J_\mu^G(\tilde{K}_G^*) + J_\mu^G(\tilde{K}_G^*) - J_\mu^D(\tilde{K}^*) \right| \\
&\leq \left| J_\pi^G(\tilde{K}_G^*) - J_\mu^G(\tilde{K}_G^*) \right| + \left| J_\mu^G(\tilde{K}_G^*) - J_\mu^D(\tilde{K}^*) \right|.
\end{aligned} \tag{123}$$

From the definition of $J_\pi^G(\tilde{K}_G^*)$, we have that $J_\pi^G(\tilde{K}_G^*) \leq J_\pi^G(\tilde{K}^*)$. Moreover, since $J_\pi^G(\tilde{K}_G^*) = J_\mu^G(\tilde{K}_G^*) + \frac{R\sigma^2\pi^2}{6}$, we have $J_\pi^G(\tilde{K}_G^*) \geq J_\mu^G(\tilde{K}_G^*)$. We thus have $J_\pi^G(\tilde{K}^*) \geq J_\pi^G(\tilde{K}_G^*) \geq J_\mu^G(\tilde{K}_G^*)$. Combining the above two results and then applying Theorem A.3, we get that

$$\left| J_\pi^G(\tilde{K}_G^*) - J_\mu^D(\tilde{K}^*) \right| \leq \left| J_\pi^G(\tilde{K}_G^*) - J_\mu^G(\tilde{K}_G^*) \right| + \left| J_\pi^G(\tilde{K}^*) - J_\mu^D(\tilde{K}^*) \right| \leq \frac{\|R\|\sigma^2\pi^2}{3}. \tag{124}$$

$\square$

We are now ready to prove Theorem 4.4, establishing weak gradient dominance of the cost $J_\pi^G(\tilde{K})$.

***Proof of Theorem 4.4.*** We first rewrite

$$\begin{aligned}
\left| J_\pi^G(\tilde{K}) - J_\pi^G(\tilde{K}_G^*) \right| = \Big| &J_\pi^G(\tilde{K}) - J_\mu^D(\tilde{K}) + J_\mu^D(\tilde{K}) + J_\mu^D(\tilde{K}_G^*) \\
&- J_\mu^D(\tilde{K}_G^*) + J_\mu^D(\tilde{K}^*) - J_\mu^D(\tilde{K}^*) - J_\pi^G(\tilde{K}_G^*) \Big|.
\end{aligned} \tag{125}$$

Rearranging terms gives

$$\begin{aligned}
\left| J_\pi^G(\tilde{K}) - J_\pi^G(\tilde{K}_G^*) \right| = \Big| &J_\pi^G(\tilde{K}) - J_\mu^D(\tilde{K}) + J_\mu^D(\tilde{K}) - J_\mu^D(\tilde{K}^*) \\
&- J_\mu^D(\tilde{K}_G^*) + J_\mu^D(\tilde{K}^*) + J_\mu^D(\tilde{K}_G^*) - J_\pi^G(\tilde{K}_G^*) \Big|.
\end{aligned} \tag{126}$$

Using the triangle inequality, we obtain

$$\begin{aligned}
\left| J_\pi^G(\tilde{K}) - J_\pi^G(\tilde{K}_G^*) \right| \leq &\left| J_\pi^G(\tilde{K}) - J_\mu^D(\tilde{K}) \right| + \left| J_\mu^D(\tilde{K}) - J_\mu^D(\tilde{K}^*) \right| \\
&+ \left| -J_\mu^D(\tilde{K}_G^*) + J_\mu^D(\tilde{K}^*) \right| + \left| J_\mu^D(\tilde{K}_G^*) - J_\pi^G(\tilde{K}_G^*) \right|.
\end{aligned} \tag{127}$$

From Theorems A.3 and A.5, we have

$$\left| J_\pi^G(\tilde{K}) - J_\pi^G(\tilde{K}_G^*) \right| \leq \frac{\|R\|\sigma^2\pi^2}{6} + \left| J_\mu^D(\tilde{K}) - J_\mu^D(\tilde{K}^*) \right| \frac{\|R\|\sigma^2\pi^2}{6} + \frac{2\|R\|\sigma^2\pi^2}{6}. \tag{128}$$

From Theorem 4.1 we can write

$$\left| J_\pi^G(\tilde{K}) - J_\pi^G(\tilde{K}_G^*) \right| \leq \frac{\|R\|\sigma^2\pi^2}{6} + \tilde{\alpha} \left\| \nabla_{\tilde{K}} J_\mu^D(\tilde{K}) \right\|^2 \frac{\|R\|\sigma^2\pi^2}{6} + \frac{2\|R\|\sigma^2\pi^2}{6}, \tag{129}$$

or, equivalently,

$$\left| J_\pi^G(\tilde{K}) - J_\pi^G(\tilde{K}_G^*) \right| \le \frac{2\|R\|\sigma^2\pi^2}{3} + \tilde{\alpha}\left\| \nabla_{\tilde{K}} J_\mu^D(\tilde{K}) \right\|^2. \tag{130}$$

Then, following Theorem A.4, we finally get

$$\left| J_\pi^G(\tilde{K}) - J_\pi^G(\tilde{K}_G^*) \right| \le \frac{2\|R\|\sigma^2\pi^2}{3} + \tilde{\alpha}\left\| \nabla_{\tilde{K}} J_\pi^G(\tilde{K}) \right\|^2. \tag{131}$$

$\square$

We are now ready to present a result on the local Lipschitzness of the LQR problem from [28], adapted for our PID control formulation. Recall that the equivalent problem in our setting is given by Proposition 2.1 with control parameter $K$, where $K$ is obtained from $\tilde{K}$ using (6).

**Lemma A.6.** *There exist constants $(\bar{m}(K), \bar{l}(K))$ for a given control parameter $K$ such that for every $K'$ satisfying $\|K' - K\| \le \bar{m}(K)$, we have*

$$\left| \nabla_K J_\pi^G(K') - \nabla_K J_\pi^G(K) \right| \le \bar{l}(K)\|K' - K\|. \tag{132}$$

*Proof.* Since we know that $J_\mu^D(K') = J(K)$, the proof follows from Theorem A.4 combined with Lemma 5 from [28]. $\square$

We establish a similar local Lipschitzness result for our cost gradient $\nabla_{\tilde{K}} J_\pi^G(\tilde{K})$ in terms of the parameter $\tilde{K}$.

**Proposition A.7.** *We have that $\sigma_{min}(M_{PI}) > 0$.*

*Proof.* From (102), we can write

$$K_P = K\frac{1}{CC^T}T_x^T C^T \tag{133}$$

$$K_I = KT_z^T. \tag{134}$$

Taking derivatives of $K_P$ and $K_I$ with respect to $K$, we get

$$\nabla_K K_P = \frac{1}{CC^T}CT_x \tag{135}$$

$$\nabla_K K_I = T_z. \tag{136}$$

Therefore, we can write $M_{PI}$ as

$$M_{PI} = \begin{bmatrix} \frac{1}{CC^T}CT_x \\ T_z \end{bmatrix}, \tag{137}$$

and, consequently,

$$M_{PI}M_{PI}^T = \begin{bmatrix} \frac{1}{CC^T}CT_x \\ T_z^T \end{bmatrix} \begin{bmatrix} \frac{1}{CC^T}(CT_x)^T & T_z \end{bmatrix} = \begin{bmatrix} \frac{1}{(CC^T)^2}CT_xT_x^TC^T & \frac{1}{CC^T}CT_xT_z^T \\ \frac{1}{CC^T}T_z(CT_x)^T & T_zT_z^T \end{bmatrix}. \tag{138}$$

From the definition of $T_x$ and $T_z$, we have

$$T_xT_x^T = I_n, T_zT_z^T = 1, T_xT_z^T = 0,$$

which yields

$$M_{PI}M_{PI}^T = \begin{bmatrix} \frac{1}{(CC^T)^2}CC^T & 0 \\ 0 & 1 \end{bmatrix}. \tag{139}$$

Finally, we establish that

$$\sigma_{min}(M_{PI}) = \sqrt{\lambda_{min}(M_{PI}M_{PI}^T)} = \sqrt{\min\{\frac{1}{CC^T}, 1\}} > 0, \tag{140}$$

where $\lambda_{min}$ represents the minimum eigenvalue. $\square$

**Corollary A.8.** *The gradient $\nabla_{\tilde{K}} J_\pi^G(\tilde{K})$ is locally Lipschitz in parameter $\tilde{K}$, that is, for all $\tilde{K}'$ such that $\left\| \tilde{K}' - \tilde{K} \right\| \leq m(\tilde{K})$, we have*

$$\left| \nabla_{\tilde{K}} J_\pi^G(\tilde{K}') - \nabla_{\tilde{K}} J_\pi^G(\tilde{K}) \right| \leq l(\tilde{K}) \left\| \tilde{K}' - \tilde{K} \right\|, \tag{141}$$

*Proof.* Consider the left hand side of (141) and write

$$h := \left| \nabla_K J_\pi^G(K') - \nabla_K J_\pi^G(K) \right|. \tag{142}$$

Then, from Theorem A.4, we know that $\nabla_K J_\pi^G(K) = \nabla_K J_\mu^D(K)$. Therefore, we get

$$h = \left| \nabla_K J_\mu^D(K') - \nabla_K J_\mu^D(K) \right|. \tag{143}$$

Applying the chain rule on $\nabla_K J_\mu^D(K)$, we get

$$\nabla_K J_\mu^D(K) = \nabla_K K_P^T \nabla_{K_P} J_\mu^D(\tilde{K}) + \nabla_K K_I^T \nabla_{K_I} J_\mu^D(\tilde{K}), \tag{144}$$

which can be rewritten as

$$\nabla_K J_\mu^D(K) = \begin{bmatrix} \nabla_K K_P^T & \nabla_K K_I^T \end{bmatrix} \begin{bmatrix} \nabla_{K_P} J_\mu^D(\tilde{K}) \\ \nabla_{K_I} J_\mu^D(\tilde{K}) \end{bmatrix}. \tag{145}$$

Note that $\nabla_{\tilde{K}} J_\mu^D(\tilde{K}) = \begin{bmatrix} \nabla_{K_P} J_\mu^D(\tilde{K}) \\ \nabla_{K_I} J_\mu^D(\tilde{K}), \end{bmatrix}$, then and therefore $\nabla_K J_\mu^D(K) = M_{PI}^T \nabla_{\tilde{K}} J_\mu^D(\tilde{K})$. Consequently, we have

$$h = \left| M_{PI}^T \nabla_{\tilde{K}} J_\mu^D(\tilde{K}') - M_{PI}^T \nabla_{\tilde{K}} J_\mu^D(\tilde{K}) \right|. \tag{146}$$

Let $\sigma_{min}(M_{PI})$ represent the minimum singular value of $M_{PI}$. Then, we can write

$$h \geq \sigma_{min}(M_{PI}^T) \left\| \nabla_{\tilde{K}} J_\mu^D(\tilde{K}') - \nabla_{\tilde{K}} J_\mu^D(\tilde{K}) \right\|. \tag{147}$$

Since $\sigma_{min}(M_{PI}) = \sigma_{min}(M_{PI}^T)$ and that $\sigma_{min}(M_{PI}) > 0$ from Proposition (A.7), we can write

$$\left\| \nabla_{\tilde{K}} J_\mu^D(\tilde{K}') - \nabla_{\tilde{K}} J_\mu^D(\tilde{K}) \right\| \leq \frac{1}{\sigma_{min}(M_{PI})} \bar{l}(K) \left\| K' - K \right\|. \tag{148}$$

Now, defining $\nabla K = K' - K$ and $\nabla \tilde{K} = \tilde{K}' - \tilde{K}$, then we can write

$$\Delta K = \begin{bmatrix} \Delta K_P C & \Delta K_I \end{bmatrix}, \tag{149}$$

or, equivalently,

$$\Delta K = S \Delta \tilde{K}. \tag{150}$$

Applying the Cauchy-Schwarz inequality, we have

$$\left\| \nabla_{\tilde{K}} J_\mu^D(\tilde{K}') - \nabla_{\tilde{K}} J_\mu^D(\tilde{K}) \right\| \leq \frac{1}{\sigma_{min}(M_{PI})} \bar{l}(K) \left\| S \right\| \left\| \tilde{K}' - \tilde{K} \right\|. \tag{151}$$

From Theorem A.4, we can finally write

$$\left| \nabla_{\tilde{K}} J_\pi^G(\tilde{K}') - \nabla_{\tilde{K}} J_\pi^G(\tilde{K}) \right| \leq l(\tilde{K}) \left\| \tilde{K}' - \tilde{K} \right\|, \tag{152}$$

where

$$l(\tilde{K}) = \frac{1}{\sigma_{min}(M_{PI})} \bar{l}(K) \left\| S \right\|, \tag{153}$$

which completes the proof. $\qquad\square$

Before moving on to proving Theorem 4.7, we present a weak gradient dominance result in parameter $\tilde{K}$, from Theorem 4.4.

**Corollary A.9** (Weak Gradient Dominance). *Suppose that Assumption 4.6 holds for the gradient* $\nabla_{\tilde{K}} J_\pi^G(\tilde{K})$. *Then the following version of weak gradient dominance holds*

$$\left| J_\pi^G(\tilde{K}) - J_\pi^G(\tilde{K}_G^*) \right| \leq \frac{2\|R\|\sigma^2\pi^2}{3} + \tilde{\alpha} \left\| \nabla_{\tilde{K}} J_\pi^G(\tilde{K}) \right\|, \text{ for all } \tilde{K} \in \tilde{\mathcal{K}}, \tag{154}$$

*where* $\tilde{\alpha} = \|\Sigma_{K^*}\| \left[ \sigma_{min}(\Sigma_K)^2 R \right]^{-1} \|M_{PI}\|^2$.

*Proof.* The above claim immediately follows from the fact that $\left\| \nabla_{\tilde{K}} J_\pi^G(\tilde{K}) \right\| \geq \left\| \nabla_{\tilde{K}} J_\pi^G(\tilde{K}) \right\|^2$ due to Assumption 4.6 and the fact that $\tilde{\alpha} > 0$. $\qquad\square$

We now provide proof of the sample complexity result of Theorem 4.7.

*Proof of Theorem 4.7.* Using the fact that $x \leq \|x\|$ and Corollary A.9, we can write

$$J_\pi^G(\tilde{K}) - J_\pi^G(\tilde{K}_G^*) \leq \tilde{\alpha} \left\| \nabla_{\tilde{K}} J_\pi^G(\tilde{K}) \right\| + \beta/3. \tag{155}$$

Define $\bar{J}_\pi^G(\tilde{K}_G^*) := J_\pi^G(\tilde{K}_G^*) + \beta/3$. Then, from (155), we have the following:

$$\frac{1}{\tilde{\alpha}} \max\{0, J_K^G - \bar{J}_\pi^G(\tilde{K}_G^*)\} \leq \left\| \nabla_{\tilde{K}} J_\pi^G(\tilde{K}) \right\|. \tag{156}$$

We note that, for time-step $t \geq 0$,

$$J_\pi^G(\tilde{K}_{t+1}) - \bar{J}_\pi^G(\tilde{K}_G^*) = J_\pi^G(\tilde{K}_{t+1}) - J_\pi^G(\tilde{K}_t) + J_\pi^G(\tilde{K}_t) - \bar{J}_\pi^G(\tilde{K}_G^*). \tag{157}$$

Using the local Lipschitz property from Corollary A.8 and the second-order Taylor expansion gives

$$J_\pi^G(\tilde{K}_{t+1}) - J_\pi^G(\tilde{K}_t) \leq \langle \nabla_{\tilde{K}} J_\pi^G(\tilde{K}_t), \tilde{K}_{t+1} - \tilde{K}_t \rangle + \frac{l(\tilde{K}_t)}{2} \|\tilde{K}_{t+1} - \tilde{K}_t\|_2^2, \tag{158}$$

and we therefore have that

$$\begin{aligned} J_\pi^G(\tilde{K}_{t+1}) - \bar{J}_\pi^G(\tilde{K}_G^*) &\leq J_\pi^G(\tilde{K}_t) - \bar{J}_\pi^G(\tilde{K}_G^*) + \langle \nabla_K J_K^G(\tilde{K}_t), \tilde{K}_{t+1} - \tilde{K}_t \rangle \\ &\quad + \frac{l(\tilde{K}_t)}{2} \|\tilde{K}_{t+1} - \tilde{K}_t\|_2^2. \end{aligned} \tag{159}$$

Recalling the update rule $\tilde{K}_{t+1} = \tilde{K}_t - \eta_t \widehat{\nabla_{\tilde{K}} J_\pi^G}(\tilde{K}_t)$, by the unbiasedness of the gradient estimator we have

$$\begin{aligned} J_\pi^G(\tilde{K}_{t+1}) - \bar{J}_\pi^G(\tilde{K}_G^*) &\leq J_\pi^G(\tilde{K}_t) - \bar{J}_\pi^G(\tilde{K}_G^*) - \eta_t \langle \nabla_{\tilde{K}} J_\pi^G(\tilde{K}_t), \widehat{\nabla_{\tilde{K}} J_\pi^G}(\tilde{K}_t) \rangle \\ &\quad + \frac{l(\tilde{K}_t)}{2} \eta_t^2 \left\| \widehat{\nabla_{\tilde{K}} J_\pi^G}(\tilde{K}_t) \right\|^2. \end{aligned} \tag{160}$$

Let $\mathcal{F}_t = \sigma(\tilde{K}_k, \eta_k; 0 \leq k \leq t)$ denote the $\sigma$-algebra generated by all randomness in the system up until time $t$, and let $\mathbb{E}_t[\cdot] = \mathbb{E}[\cdot|\mathcal{F}_{t-1}]$. Taking expectations over the foregoing yields

$$\begin{aligned} \mathbb{E}_t \left[ J_\pi^G(\tilde{K}_{t+1}) - \bar{J}_\pi^G(\tilde{K}_G^*) \right] &\leq \mathbb{E}_t \left[ J_\pi^G(\tilde{K}_t) - \bar{J}_\pi^G(\tilde{K}_G^*) - \eta_t \langle \nabla_{\tilde{K}} J_\pi^G(\tilde{K}_t), \widehat{\nabla_{\tilde{K}} J_\pi^G}(\tilde{K}_t) \rangle \right. \\ &\quad \left. + \frac{l(\tilde{K}_t)}{2} \eta_t^2 \left\| \widehat{\nabla_{\tilde{K}} J_\pi^G}(\tilde{K}_t) \right\|^2 \right]. \end{aligned} \tag{161}$$

Given $\mathbb{E}_t[\widehat{\nabla_{\tilde{K}}} J_\pi^G(\tilde{K}_t)] = \nabla_{\tilde{K}} J_\pi^G(\tilde{K}_t)$ from the fact the our estimator is unbiased, and recalling the definition of the variance of a random variable, we can write the following

$$
\begin{aligned}
\mathbb{E}_t\left[J_\pi^G(\tilde{K}_{t+1}) - \bar{J}_\pi^G(\tilde{K}_G^*)\right] &\le J_\pi^G(\tilde{K}_t) - \bar{J}_\pi^G(\tilde{K}_G^*) - \eta_t\langle\nabla_{\tilde{K}} J_\pi^G(\tilde{K}_t), \nabla_{\tilde{K}} J_\pi^G(\tilde{K}_t)\rangle \\
&\quad + \frac{l(\tilde{K}_t)}{2}\eta_t^2 Var(\widehat{\nabla_{\tilde{K}}} J_\pi^G(\tilde{K}_t)) + \frac{l(\tilde{K}_t)}{2}\eta_t^2 (\mathbb{E}_t\left[\left\|\widehat{\nabla_{\tilde{K}}} J_\pi^G(\tilde{K}_t)\right\|\right])^2, \\
&= J_\pi^G(\tilde{K}_t) - \bar{J}_\pi^G(\tilde{K}_G^*) - \eta_t\left\|\nabla_{\tilde{K}} J_\pi^G(\tilde{K}_t)\right\|^2 \\
&\quad + \frac{l(\tilde{K}_t)}{2}\eta_t^2 Var(\widehat{\nabla_{\tilde{K}}} J_\pi^G(\tilde{K}_t)) + \frac{l(\tilde{K}_t)}{2}\eta_t^2 \left\|\nabla_{\tilde{K}} J_\pi^G(\tilde{K}_t)\right\|^2, \\
&= J_\pi^G(\tilde{K}_t) - \bar{J}_\pi^G(\tilde{K}_G^*) - \eta_t\left(1 - \frac{l(\tilde{K}_t)}{2}\eta_t\right)\left\|\nabla_{\tilde{K}} J_\pi^G(\tilde{K}_t)\right\|^2 \\
&\quad + \frac{l(\tilde{K}_t)}{2}\eta_t^2 Var(\widehat{\nabla_{\tilde{K}}} J_\pi^G(\tilde{K}_t)).
\end{aligned}
\tag{162}
$$

Using Assumption 4.5, we finally obtain

$$
\begin{aligned}
\mathbb{E}_t\left[J_\pi^G(\tilde{K}_{t+1}) - \bar{J}_\pi^G(\tilde{K}_G^*)\right] &\le J_\pi^G(\tilde{K}_t) - \bar{J}_\pi^G(\tilde{K}_G^*) - \eta_t\left(1 - \frac{l(\tilde{K}_t)}{2}\eta_t\right)\left\|\nabla_{\tilde{K}} J_\pi^G(\tilde{K}_t)\right\|^2 \\
&\quad + \frac{l(\tilde{K}_t)}{2}\eta_t^2\frac{\bar{V}}{N}.
\end{aligned}
\tag{163}
$$

For a sequence of Lipschitz constants $\{l(K_i)\}$, suppose that the largest Lipschitz constant is given by $l_{mx}$. Further, suppose that we pick a step size $\eta_t \le \frac{1}{l_{mx}}$; then it follows that $(1 - \frac{l_{mx}}{2}\eta_t \ge \frac{1}{2})$. We can then re-write the above inequality as follows

$$
\mathbb{E}_t\left[J_\pi^G(\tilde{K}_{t+1}) - \bar{J}_\pi^G(\tilde{K}_G^*)\right] \le J_\pi^G(\tilde{K}_t) - \bar{J}_\pi^G(\tilde{K}_G^*) - \frac{\eta_t}{2}\left\|\nabla_{\tilde{K}} J_\pi^G(\tilde{K}_t)\right\|^2 + \frac{l(\tilde{K}_t)}{2}\eta_t^2\frac{\bar{V}}{N}.
\tag{164}
$$

At this point, we can now follow the proof of Theorem 6.1 in [33] with $L_2 = l_{mx}$ to obtain the bound

$$
\mathbb{E}\left[J_\pi^G(\tilde{K}_T)\right] - J_\pi^G(\tilde{K}_G^*) \le \beta/3 + c_T\left(1 - \sqrt{\frac{\eta^3 l_{mx}^T \bar{V}}{4\tilde{\alpha}^2 N}}\right)^T + \sqrt{\frac{l_{mx}^T \bar{V}\eta\tilde{\alpha}^2}{N}},
\tag{165}
$$

where we note that, for finite $T$, we can replace the $l_{mx}$ by $l_{mx}^T$. From Theorem A.5, we have

$$
\left|J_\pi^G(\tilde{K}_G^*) - J_\mu^D(\tilde{K}^*)\right| \le \frac{2\|R\|\sigma^2\pi^2}{6} = 2\beta/3.
\tag{166}
$$

Note that, by (117), we have $J_\pi^G(\tilde{K}_G^*) = J_\mu^D(\tilde{K}_G^*) + \frac{R\sigma^2\pi^2}{6}$. Furthermore, since $J_\mu^D(\tilde{K}_G^*) \ge J_\mu^D(\tilde{K}^*)$, we obtain

$$
J_\pi^G(\tilde{K}_G^*) \ge J_\mu^D(\tilde{K}^*) + \frac{R\sigma^2\pi^2}{6}.
\tag{167}
$$

In other words, we have that $J_\mu^D(\tilde{K}^*)$ is the global minimum. Noisily perturbing the function $J_\mu^D(\tilde{K})$ to yield $J_\pi^G(\tilde{K})$ leads to a minimum that satisfies $J_\mu^D(\tilde{K}^*) \le J_\pi^G(\tilde{K}_G^*)$, or, equivalently, $J_\pi^G(\tilde{K}_G^*) - J_\mu^D(\tilde{K}^*) \ge 0$. From the fact that $x \le \|x\|$, this implies

$$
J_\pi^G(\tilde{K}_G^*) - J_\mu^D(\tilde{K}^*) \le 2\beta/3,
\tag{168}
$$

i.e., that

$$
-J_\pi^G(\tilde{K}_G^*) \ge -J_\mu^D(\tilde{K}^*) - 2\beta/3.
\tag{169}
$$

We therefore finally obtain

$$
\mathbb{E}\left[J_\pi^G(\tilde{K}_T)\right] - J_\mu^D(\tilde{K}^*) \le \beta + c_T\left(1 - \sqrt{\frac{\eta^3 l_{mx}^T \bar{V}}{4\tilde{\alpha}^2 N}}\right)^T + \sqrt{\frac{l_{mx}^T \bar{V}\eta\tilde{\alpha}^2}{N}}.
$$

The remainder of Theorem 4.7 now follows in a straightforward manner from the rest of the proof of [33, Theorem 6.1]. $\qquad\square$

## A.4 Experiment Details

We note that both the environments `rea` and `lah` have scalar input and output spaces. In our experiments, we set $R = Y = 1.0$, where matrices $R$ and $Y$ are as in (1). Further, the cost accumulated over each length-$N$ rollout, at time $t$, is given by

$$-\sum_{i=t}^{t+N-1} \left( g_i^T Q g_i + u_i^T R u_i \right). \tag{170}$$

For the experiments generating reward curves in Figures 1, we ran 100 trials of our proposed algorithms and 4 trials of the PPO algorithm. The simulation hyperparameters for our experiments are provided in Tables 1, 2, 3 and 4. Finally, we ran our experiments on a Windows laptop, configured with a 6-core i7-8750H, 2.20GHz CPU, an NVIDIA GeForce RTX 2070 GPU, and 32GB RAM.

Table 1: Simulation Parameters for PG4PID

| Parameters | Values |
|---|---|
| No. of episodes $T$ | 10 |
| Rollout length $N$ | 100 |
| Step sizes $\{\alpha_P, \alpha_I, \alpha_D\}$ | 0.0001 |
| Tolerance $\epsilon$ | 0.001 |
| Sampling time $\tau$ | 0.1 sec |

Table 2: Simulation Parameters for PG4PI

| Parameters | Values |
|---|---|
| No. of episodes $T$ | 10 |
| Rollout length $N$ | 100 |
| Step sizes $\{\alpha_P, \alpha_I, \alpha_D\}$ | 0.0001 |
| Noise variance $\sigma$ | 0.1 |
| Tolerance $\epsilon$ | 0.001 |
| Sampling time $\tau$ | 0.1 sec |

Table 3: Simulation Parameters for Model-free LQR

| Parameters | Values |
|---|---|
| No. of episodes $T$ | 10 |
| Rollout length $N$ | 100 |
| Step size $\alpha$ | 0.0001 |
| Tolerance $\epsilon$ | 0.001 |
| Sampling time $\tau$ | 0.1 sec |

Table 4: Simulation Parameters for PPO

| | |
|---|---|
| Policy learning rate | 0.0003 |
| Palue learning rate | 0.0003 |
| Entropy coefficient | 0.0 |
| Clip range | 0.2 |
| Weight decay | 0.0 |
| Layer size | 2 |
| Batch size | 2 |
| Buffer size | 60000 |
| Number of epochs | 10 |
| Rollout length | 8 |
| Discount factor | 0.99 |

### A.4.1 Validation on `rea` and `lah`

We present the state, input tracking, and tracking error trajectories for our experiments on the Chemical Reactor (`rea`) and the LA University Hospital (`lah`) `controlgym` environments. Recall that we consider the unit step tracking task which entails two subtasks: i) system output tracking unit step input, and ii) stabilizing the closed loop system. The step tracking performance is shown in Figures 3a and 3b with the model-based PID Algorithm PG4PID, and Figures 3c and 3d with the model-free PI Algorithm PG4PI. We also plot corresponding trajectories for the tracking error (deviation from reference step input) in Figures 3e and 3f with PG4PID, and Figures 3g and 3h with PG4PI. Finally, we illustrate the efficacy of our model-based and model-free algorithms for the task of system stabilization. The corresponding state trajectories are illustrated in Figures 4a and 4b with PG4PID, and Figures 4c and 4d with PG4PI.

### A.4.2 Ablation Studies

We carry out ablation studies on the impact of the variance parameter $\sigma$ on the convergence of our Model-free PI Algorithm, PG4PI, on both the Chemical Reactor and LA Hospital environments. The results are presented in Figure 5. We find that varying the $\sigma$ does not have a significant impact on algorithm performance for small $\sigma$ (between 0.0001-2); however, some performance degradation occurs for larger variances ($\sigma = 3$). In general, $\sigma$ is chosen to be small enough that suitable gradient dominance and convergence results can still be recovered. In practice, the choice of $\sigma$ comes down

to ensuring numerical stability, and we find from our ablation study that Algorithm 2 is stable for a wide range of $\sigma$ values.

### A.4.3  Benchmarking vs. PG based LQR

We present the results of our benchmarking experiment where we implement both a model-free (zero-order optimization method) PG method and model-based (using exact gradient expressions) method for tuning LQR control policies from [18] in Figures 6 and 7 respectively. In model-based LQR, we use estimates of matrices $(A, B)$ to compute the policy gradient, while we use their Algorithm 1 in [18] to estimate the gradient in the model-free method. The experimental parameters for model-free LQR are presented in Table 3. We note that our proposed model-free Algorithm 2 follows a first-order optimization method, that our model-free PID control policy outperforms both model-based and model-free LQR control policies in both tracking behavior and robustness to model errors, as discussed in Section 5.

### A.5  Broader Impacts

This work is primarily theoretical and pertains to the development of policy gradient based algorithms to design PID control policies, and does not have any direct societal impact.

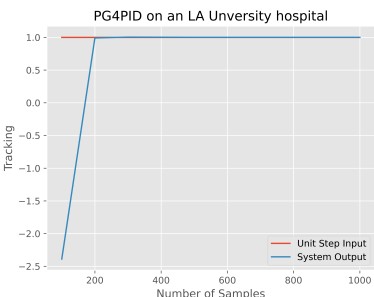

(a) Tracking performance for unit step reference in `lah` environment with PG4PID.

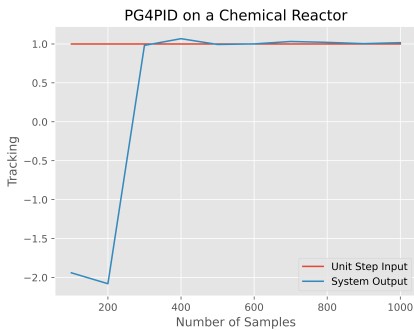

(b) Tracking performance for unit step reference in `rea` environment with PG4PID.

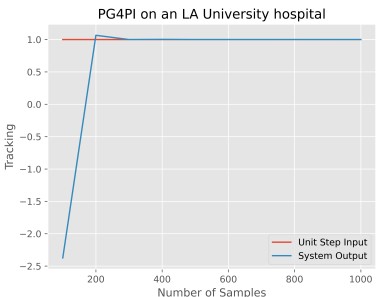

(c) Tracking performance for unit step reference in `lah` environment with PG4PI.

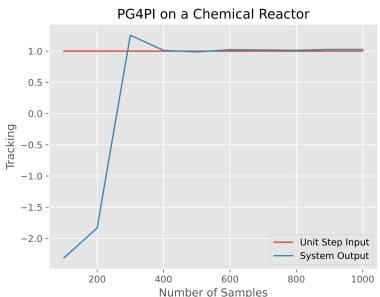

(d) Tracking performance for unit step reference in `rea` environment with PG4PI.

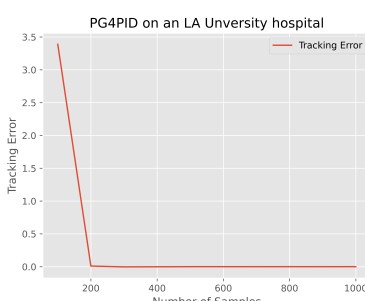

(e) Tracking error with unit step reference in `lah` with PG4PID.

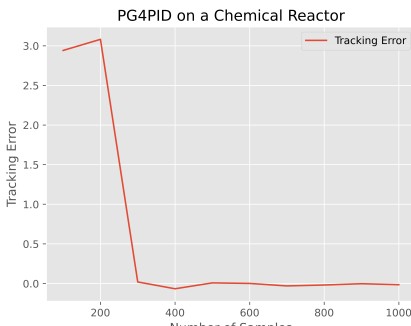

(f) Tracking error with unit step reference in `rea` with PG4PID.

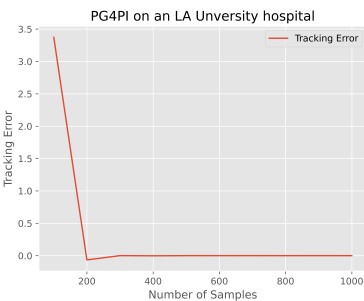

(g) Tracking error with unit step reference in `lah` with PG4PI.

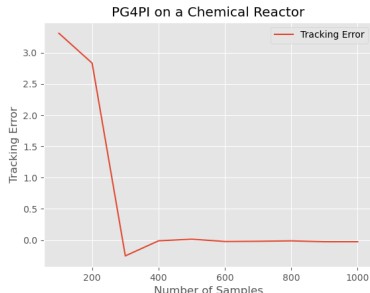

(h) Tracking error with unit step reference in `rea` with PG4PI.

Figure 3: Output trajectories and tracking errors in response to unit step reference input signal, and illustrating perfect tracking with PG4PID and PG4PI.

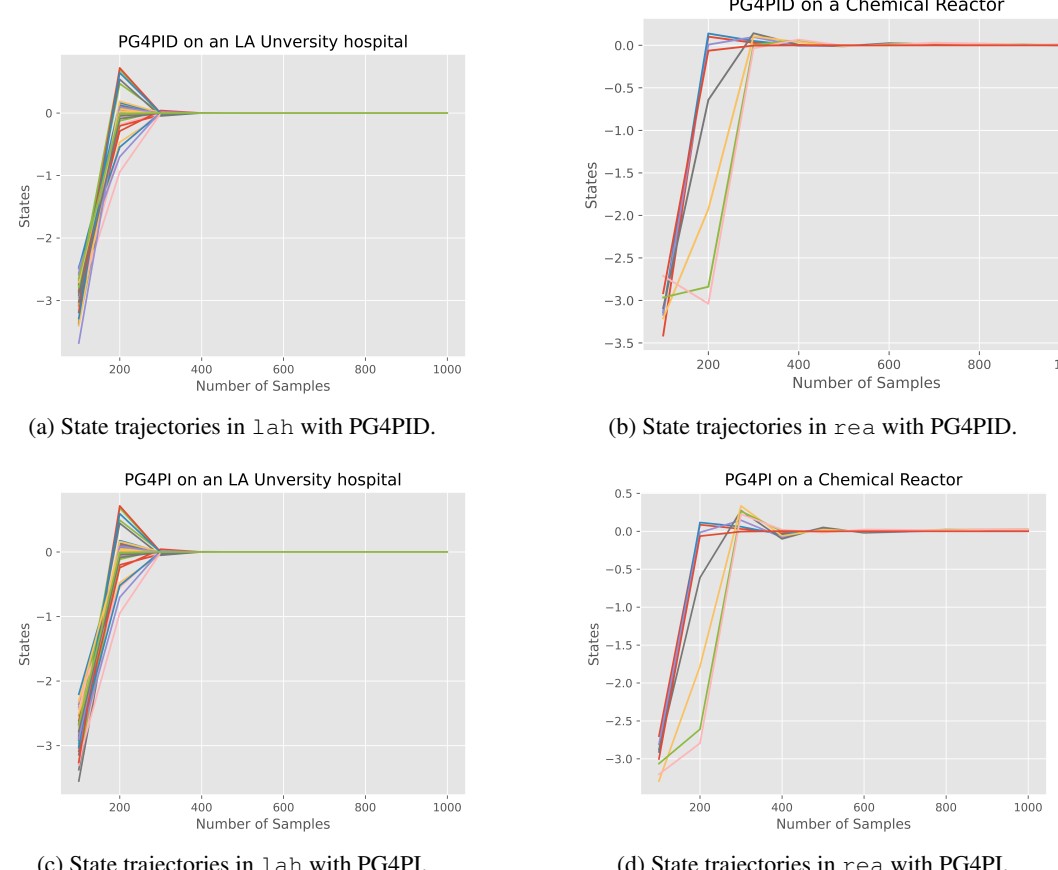

(a) State trajectories in `lah` with PG4PID.

(b) State trajectories in `rea` with PG4PID.

(c) State trajectories in `lah` with PG4PI.

(d) State trajectories in `rea` with PG4PI.

Figure 4: State trajectories for the `lah` and `rea` environments illustrating stabilization with PG4PID and PG4PI.

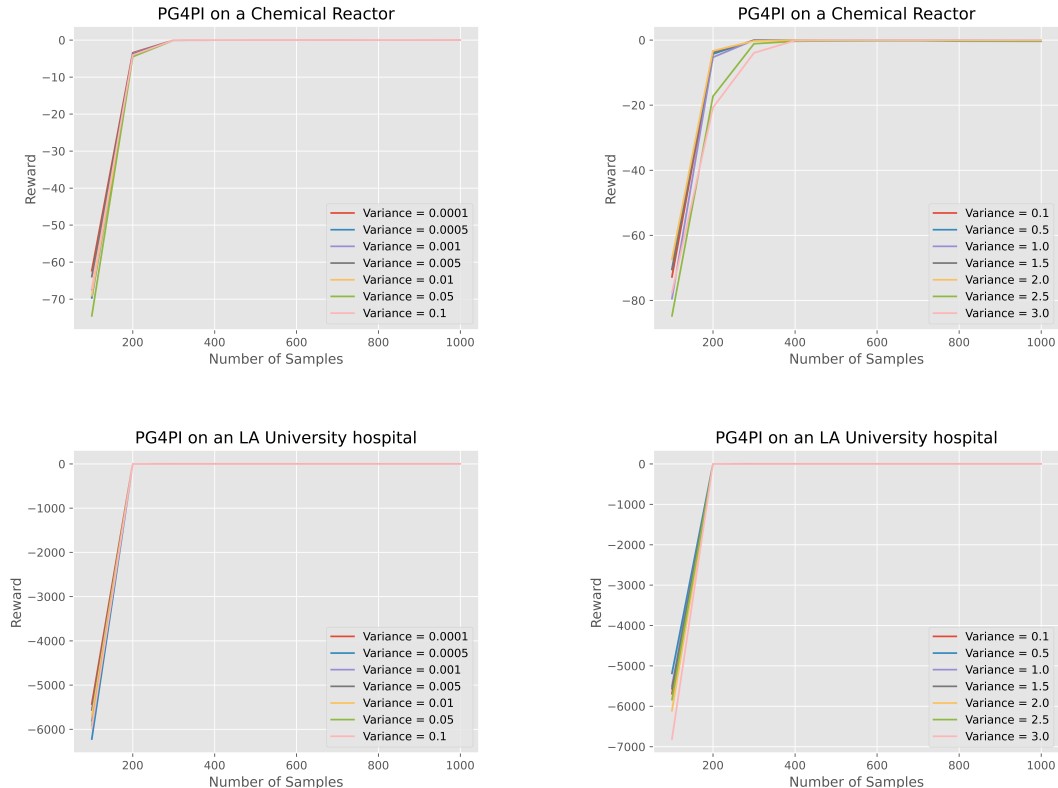

Figure 5: Reward curves for the PG4PI Algorithm on the 8-dimensional Chemical Reactor and the on the 48-dimensional LA University Hospital environments. Plots contain two sets of experiments, run for a smaller set and a larger set of variances $\sigma$. While smaller variances ($\sigma < 2$) have negligible effect on the convergence rates, slower convergence is observed for some larger variances ($\sigma = 3$).

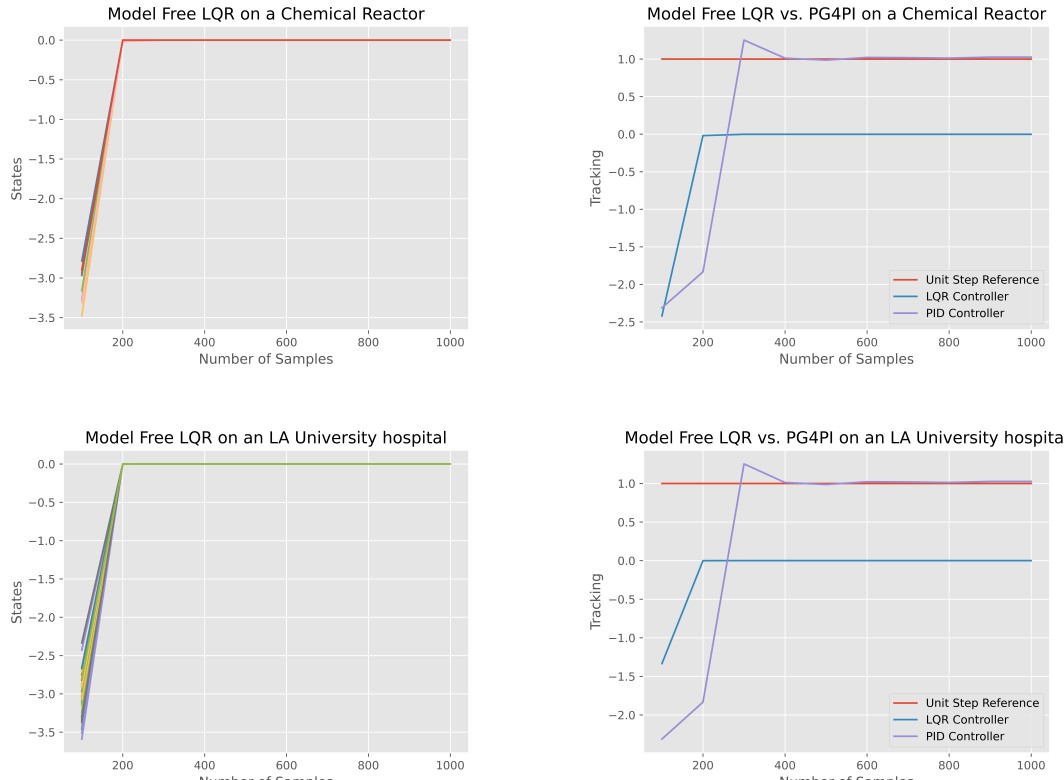

Figure 6: State stabilization trajectories and tracking performance for a unit step reference with model-free PG with LQR policy and model-free PG4PI on the Chemical Reactor and LA University Hospital environments. While the Model-free LQR controller stabilizes the states for both the environments, there is a non-zero steady state tracking error for the unit step input, while the model-free PI controller achieves both stabilization and perfect tracking with zero steady-state error.

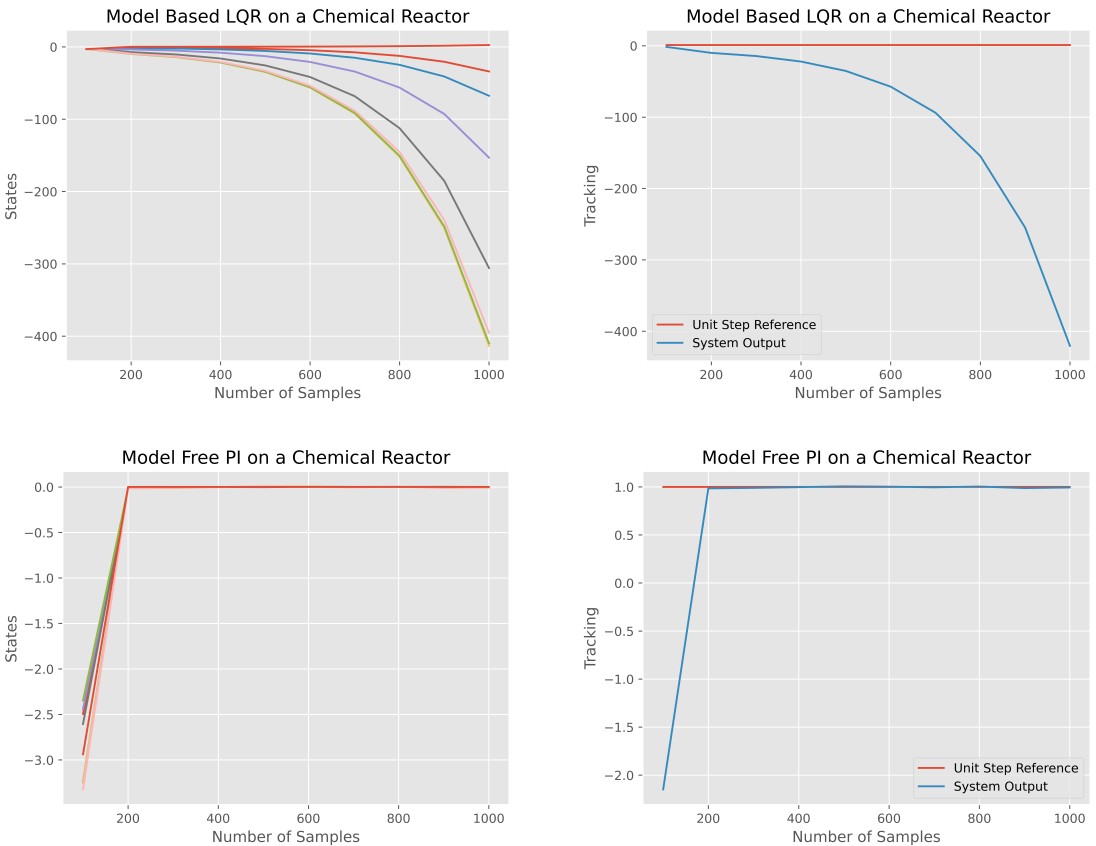

Figure 7: Comparison of model-based PG with LQR policy and model-free PG4PI with small model error for the Chemical Reactor environment. LQR is highly fragile and exhibits instability with a model error of $\|\Delta A\| = 0.05$ even with a model-based implementation, while our model-free PI controller achieves stabilization and perfect tracking with zero steady-state error.

