# OpenReview forum: "Globally Optimal Policy Gradient Algorithms for Reinforcement Learning with PID Control Policies"
_NeurIPS.cc/2025/Conference — NeurIPS 2025 poster_

### Official Review · Reviewer_bcra · 2025-06-29

**Clarity:** 2
**Significance:** 1
**Originality:** 2
**Rating:** 2
**Confidence:** 3

**Summary:**

This paper proposes policy gradient algorithms for reinforcement learning with PID-parameterized control policies. The method leverages the practical effectiveness of PID controllers while providing theoretical guarantees on convergence and global optimality in both model-based and model-free settings.

**Questions:**

1. Why are the gradients derived from Equation 3 instead of directly from Equation 1? Clarifying this design choice would help in understanding the derivation and optimization approach.

**Ethical Concerns:**

["NO or VERY MINOR ethics concerns only"]

**Final Justification:**

Thanks for the rebuttal. I am still concerned about the novelty of the method, where many works used RL to tune the parameters of classic controllers. And experimental results are also limited. I will keep my score.

**Limitations:**

Yes

**Quality:**

2

**Strengths And Weaknesses:**

**Strengths:**

- The mathematical notation is consistent and clearly presented, which aids understanding of the theoretical analysis.

**Weaknesses:**

- The main limitation lies in the novelty of the contribution. Prior work has already explored reinforcement learning for fine-tuning structured controllers such as MPC and PID [1, 2, 3]. The paper would benefit from a clearer distinction between its contributions and those of existing methods.
- The experimental results are difficult to interpret. In Figure 1, all curves appear very close together, making it hard to evaluate relative performance. Additionally, many results are reported without uncertainty estimates (e.g., error bars), which weakens empirical claims.

[1] Gros, Sébastien and Mario Zanon. “Data-Driven Economic NMPC Using Reinforcement Learning.” *IEEE Transactions on Automatic Control* 65 (2019): 636-648.
 [2] Hardt, Moritz et al. “Gradient Descent Learns Linear Dynamical Systems.” *arXiv preprint* arXiv:1609.05191 (2016).
 [3] Westenbroek, Tyler et al. “Feedback Linearization for Unknown Systems via Reinforcement Learning.” *arXiv preprint* arXiv:1910.13272 (2019).

---

> ### Author Rebuttal · Authors · 2025-07-30
>
> Thank you for your time and efforts in reviewing this work.
>
> **a) On novelty and significance of our work:**
> > The main limitation lies in the novelty of the contribution. Prior work has already explored reinforcement learning for fine-tuning structured controllers such as MPC and PID [1, 2, 3]. The paper would benefit from a clearer distinction between its contributions and those of existing methods.
> > [1] Gros, Sébastien and Mario Zanon. “Data-Driven Economic NMPC Using Reinforcement Learning.” IEEE Transactions on Automatic Control 65 (2019): 636-648.
> > [2] Hardt, Moritz et al. “Gradient Descent Learns Linear Dynamical Systems.” arXiv preprint arXiv:1609.05191 (2016).
> > [3] Westenbroek, Tyler et al. “Feedback Linearization for Unknown Systems via Reinforcement Learning.” arXiv preprint arXiv:1910.13272 (2019).
>
> **Response:** We respectfully disagree: to the best of our knowledge, there is no existing literature on policy gradient algorithms for ***PID control with theoretical guarantees***. We emphasize that none of the works that the reviewer cites deal with the PID parameterization. More precisely, [2] focuses on system identification and does not consider a control design problem, [3] develops methods to linearize a nonlinear system through feedback, and [1] focuses does not explicitly tune structured policies (only providing LQR policy as case study), and requires assumptions such as strict dissipativity, which do not hold for the general class of problems we consider.
>
> In fact, most prior works in RL-based tuning of structured controllers have focused on the widely studied LQR architecture [4]. Additionally, while MPC provides a powerful model-free control design framework, it does not yield structured policies in general. Meanwhile, PID controllers are **ubiquitous in most practical applications** including robotics, aerospace, and chemical process control, with >90% of controllers deployed in industry being PID controllers [5]. Yet, to the best of our knowledge, there is no work that rigorously addresses the PID policy parameterization in RL with theoretical guarantees.
>
> Importantly, our **contribution goes well beyond applying existing RL methods to tune a structured controller**. Rather, we develop a **complete policy gradient theory with provable guarantees on convergence and global optimality** for this widely useful class of policies.
>
> Specifically, we derive novel policy gradient expressions for the PID control problem in both the model-based and model-free settings (Theorem 3.1 & Corollary 3.3). In contrast to existing approaches, which use zero-order methods to estimate the gradients, these exact gradient expressions allow us to design first-order PG algorithms that are also broadly applicable to general LQR problems, which are a special case of our framework. We also establish gradient dominance of this problem (Theorems 4.1 & 4.4), which allows us to provide proofs of optimality, convergence, and sample complexity (Theorems 4.2, 4.7). In the model-free setting, we propose a novel policy gradient approach (Algorithm 2) where the key innovation stems from the fact that we are provably learning the optimal parameters of a deterministic PID controller while updating the stochastic policy of eq. (15); here (i) the PID gradients derived in Section 3 are new and necessary, and (ii) the guarantees of Section 4 required substantial innovation over existing analyses.
>
> We also demonstrate through experiments that **our algorithms outperform direct application of RL to tune the structured controller** (see benchmark on tuning PID controllers with PPO Section 5).
>
> Taken together, these are ***nontrivial, novel contributions*** that provide provably convergent and optimal PG algorithms tailored to the important and widely useful class of PID control policies.
>
> **b) On experimental results:**
> >The experimental results are difficult to interpret. In Figure 1, all curves appear very close together, making it hard to evaluate relative performance. Additionally, many results are reported without uncertainty estimates (e.g., error bars), which weakens empirical claims.
>
> **Response:** Thank you for this comment. All reward curves in Figure 1 are shown with error bands, but we agree that they may be hard to discern at the current scale. We will increase the size of Fig. 1 so that all reward curves and error bands are clearly discernible. While it is challenging to incorporate error bands in the state trajectory plots due to high-dimensional state spaces of our environments (8- and 48-dimensional respectively), we will include error bands in our tracking plots in Figure 2 in the final version.
>
> **c) On policy gradient derivation:**
> > Why are the gradients derived from Equation 3 instead of directly from Equation 1? Clarifying this design choice would help in understanding the derivation and optimization approach.
>
> **Response:** We appreciate this question, though we would like to clarify that this is precisely the key technical challenge addressed by our paper. As discussed in lines 131-140, it is not feasible to directly derive gradient expressions from Equation 1 due to the fact that though $K$ in (6) is completely determined given the PID parameters $\tilde K$, there does not necessarily exist a feasible decomposition of $K$ in terms of $\tilde K$. For this reason, it is not possible to use existing methods to derive policy gradient expressions directly from the problem in Equation (1), and then translate the resulting controller into a feasible PID controller. Therefore, it is necessary to perform PG descent directly in the PID parameters. To achieve this, we derive gradient expressions in the PID parameters for the reformulated problem in Equation (3), as detailed in Section 2.
>
> We hope that we have been able to address your concerns. We look forward to any additional questions you may have, and hope that you will consider raising your evaluation score of our work accordingly.
>
> [1] Gros, Sébastien and Mario Zanon. “Data-Driven Economic NMPC Using Reinforcement Learning.” IEEE Transactions on Automatic Control 65 (2019): 636-648
> [2] Hardt, Moritz et al. “Gradient Descent Learns Linear Dynamical Systems.” arXiv preprint arXiv:1609.05191 (2016)
> [3] Westenbroek, Tyler et al. “Feedback Linearization for Unknown Systems via Reinforcement Learning.” arXiv preprint arXiv:1910.13272 (2019)
> [4] Hu, B., Zhang, K., Li, N., Mesbahi, M., Fazel, M. and Başar, T., 2023. Toward a theoretical foundation of policy optimization for learning control policies. Annual Review of Control, Robotics, and Autonomous Systems, 6(1), pp.123-158
> [5] Hägglund, T. and Guzmán, J.L., 2024. Give us PID controllers and we can control the world. IFAC-PapersOnLine, 58(7), pp.103-108

---

> > ### Comment · Reviewer_bcra · 2025-08-05
> >
> > Thank the authors for the rebuttal.
> >
> > I have a few further questions. The policy in the PPO experiment generates the PID parameters K_P, K_I and K_D,  is that correct? Is the policy updated the same as Equation 16, or is it updated by some learned value functions in a temporal difference manner?  Could you inform whether the inclusion of this RL structure influences the convergence guarantee? Besides, it would be interesting to compare with the direct policy (without PID parameterisation) trained with PPO.

---

> > > ### Author Response · Authors · 2025-08-06
> > >
> > > Thank you for taking the time to engage with our response! We address each of your questions below.
> > >
> > > **Question 1**
> > > >The policy in the PPO experiment generates the PID parameters K_P, K_I and K_D, is that correct?
> > >
> > > **Response:** Yes, the PPO algorithm generates the $K_P, K_I$ and $K_D$ parameters in our experiment.
> > >
> > > **Question 2**
> > > >Is the policy updated the same as Equation 16, or is it updated by some learned value functions in a temporal difference manner?
> > >
> > > **Response:** To compare our algorithm with the standard PPO algorithm, we applied the PPO implementation from StableBaselines3 (lines 316-317 of our paper). This implementation learns a critic using a TD-style loss, which is then used to form the PPO policy loss (see Eq. (9) in [Schulman et al., "Proximal Policy Gradient Algorithms"]) that is subsequently optimized. The PID parameters generated from PPO are then used to construct the controller using Eq. (2). The StableBaselines3 PPO implementation we used does not use eq. (16) from our paper.
> > >
> > >
> > > **Question 3**
> > > >Could you inform whether the inclusion of this RL structure influences the convergence guarantee?
> > >
> > > **Response:** We note that PPO does not offer global convergence guarantees, regardless of the controller structure or parameterization. Therefore, using PPO to tune PID parameters does not yield any formal guarantees on convergence or optimality. In contrast, our algorithms utilize exact analytical expressions for the policy gradients in terms of the policy parameters derived in Theorem 3.1 and Corollary 3.3, which allows us to provide global convergence and optimality guarantees. Further, the high-dimensional state spaces of our environments makes PPO converge much slower than our proposed algorithms (Section 5).
> > >
> > > **Suggestion 4**
> > > >Besides, it would be interesting to compare with the direct policy (without PID parameterisation) trained with PPO.
> > >
> > > **Response:** We began by implementing the baseline PPO algorithm in the ControlGym environment (found at 'controlgym/controlgym/controllers/ppo.py' in the ControlGym repository [Reference [49] in our paper]), and found that this implementation in fact destabilizes the 'rea' environment. This motivated comparison with a more standard PPO implementation, wherein we chose the StableBaselines3 PPO algorithm. Here, the convergence of the standard PPO algorithm for both the regulation (driving the state to zero) and tracking (following a reference input) tasks requires a significantly larger number of samples due to the high-dimensional search space of our environment.
> > >
> > > We initially limited the results presented in our experiment section to structured policies (PPO parameterized by PID) to ensure a fair comparison in the benchmark experiments, but are happy to include results of both the ControlGym PPO implementation and StableBaselines3 PPO implementation (without PID parameterization) in the final version of our paper.
> > >
> > > We hope that these clarifications are helpful and are happy to address any further questions you may have!

---

### Official Review · Reviewer_1Cp5 · 2025-06-30

**Clarity:** 4
**Significance:** 3
**Originality:** 3
**Rating:** 5
**Confidence:** 3

**Summary:**

This paper bridges the gap between reinforcement learning and PID control by introducing policy gradient (PG) methods tailored for PID architectures. While PG methods have strong theoretical guarantees in classical control problems, their application to PID—despite its widespread use—has been largely heuristic. The authors propose an optimization-based framework for PID control, derive exact gradient expressions, and develop two key algorithms: a model-based method for full PID control, and a model-free method specifically for PI control. The model-based approach leverages system knowledge to achieve global convergence with linear rates, while the model-free algorithm provably learns optimal PI parameters using stochastic policies, offering global guarantees under weaker assumptions.

**Questions:**

1. Are Assumptions 4.3, 4.5, and 4.6 likely to be satisfied in practical implementations?

2. When extending to constrained optimization settings, what needs to be achieved, and what is the main obstacle?

**Ethical Concerns:**

["NO or VERY MINOR ethics concerns only"]

**Final Justification:**

The authors provided a thorough and thoughtful rebuttal. Their clarifications regarding technical assumptions, visualization improvements, and the restructuring of experimental results were clear, well-motivated, and addressed my key concerns. The response demonstrated a strong command of both the theoretical underpinnings and practical considerations of their proposed framework.

- Resolved Issues:
1. Assumptions 4.3, 4.5, and 4.6 were justified convincingly as standard or practically relaxable in implementation, with supporting ablation evidence and reference to widely used techniques such as gradient normalization.

2. The authors gave a thoughtful and technically grounded discussion on the challenges of extending the framework to constrained settings, demonstrating a clear awareness of open research directions.

3. They committed to improving visual clarity (e.g., Figure 1), correcting minor errors (e.g., Equation (3)), and moving important experiments (such as LQR and stability analysis) into the main paper, which will substantively enhance the readability and completeness of the empirical section.

- Remaining Suggestions:
1. While space limitations are understandable, the planned integration of empirical results into the main text is an important improvement, and I encourage the authors to ensure that future readers benefit from these clarifications.

2. The theoretical results are a clear strength of the paper, and the planned discussion on constrained settings—even if brief—will make the work more forward-looking and comprehensive.

- Overall Evaluation:

The rebuttal not only addressed the main concerns but also improved my overall confidence in the paper’s contribution and presentation. Based on the clarity, responsiveness, and technical depth of the rebuttal, I have raised my score accordingly. I look forward to seeing the revised version.

**Limitations:**

yes

**Quality:**

3

**Strengths And Weaknesses:**

**Strengths:**

1. This paper is written in a very elegant and fluent manner, making it easy to understand. It introduces the problem through a clear presentation of the background and proposes a solution supported by solid theoretical analysis.

2. The paper establishes global optimality, convergence rates, and sample complexity guarantees for policy gradient methods applied to PID and PI control. Specifically, it proves gradient dominance for the model-based PID setting and derives weak gradient dominance and sample complexity bounds for the model-free PI algorithm.

3. The paper compares its methods with both PG-based LQR algorithms and PPO-based approaches, effectively resolving concerns regarding performance and practical applicability.

**Weaknesses:**

1. The visualization in the experimental section could be improved. For example, in Figure 1, it is recommended to truncate the curves to a smaller number of samples (e.g., the first 500) to better illustrate the dynamic changes. Currently, the mean trajectories are almost completely overlapped with the 1-SD bands, making it difficult to distinguish between different methods and limiting the readability and comparability of the results.

2. Since both PPO and LQR are included as baselines in the experimental section, it would be better to move the comparison with LQR into the main text to facilitate a clearer comparison of the results.

3. If possible, the stability experiments related to Figure 6 would be better presented in the main text to more intuitively demonstrate the results. Currently, the experimental section takes up less than one page and feels somewhat flat.

Minor:
- Second equation in equation (3) should be: $g_{t+1} = \overline{A}g_t + \overline{B}u_t$

---

> ### Author Rebuttal · Authors · 2025-07-30
>
> Thank you for your time, efforts, and feedback, and for the encouraging evaluation of our work! We address your comments below.
>
> **a) On Assumptions 4.3, 4.5, and 4.6:**
>
> > Are Assumptions 4.3, 4.5, and 4.6 likely to be satisfied in practical implementations?
>
> Thank you for bringing up this question on the practicality of our assumptions and allowing us an opportunity to clarify. We address each assumption individually below, highlighting how all of them can be relaxed in practice.
>
> *(i) Assumption 4.3:*
> This is akin to technical assumptions on decreasing step size assumption that are common across most gradient-based algorithms including stochastic gradient descent (SGD). In practice, we can  select any decreasing sequence $\{\sigma_t\}$ as an alternative. In fact, in our experiments, we observed that all our experiments converged even with a *constant* $\sigma_t$, that is $\sigma_t=\sigma, \forall t \geq 0$. We also demonstrate through ablation studies that our algorithms retain performance and convergent behavior for a wide range of $\sigma$ (Fig. 4) Therefore, we believe that this assumption can be relaxed in practical implementations.
>
> *(ii) Assumption 4.5:*
> Theoretically, as mentioned in lines 290-292, this assumption is expected to hold in general, with reasonable conditions on sigma (small enough, as shown in our ablation studies in Fig. 4) and a differentiable cost function with reasonable Lipschitz bounds. In practice, we can obtain gradient estimates with bounded variance by choosing a sufficient rollout length and obtaining a sufficient number of trajectory samples, which is a common strategy to decrease variance in all RL implementations.
>
> *(iii) Assumption 4.6:*
> This assumption is purely for technical convenience, as mentioned in lines 292-293. In practice, we can relax this assumption by gradient normalization whenever the gradient norm exceeds unity. This is a common strategy in most learning algorithms, including neural network training and policy gradient methods, to avoid exploding gradients [1]. However, in our experiments, we observed that our algorithms showed convergence even without gradient normalization, suggesting that this assumption can also be relaxed in practical implementations.
>
> [1] Chen, Z., Badrinarayanan, V., Lee, C.Y. and Rabinovich, A., 2018, July. Gradnorm: Gradient normalization for adaptive loss balancing in deep multitask networks. In International conference on machine learning (pp. 794-803). PMLR.
>
> **b) On extension to constrained optimization:**
>
> > When extending to constrained optimization settings, what needs to be achieved, and what is the main obstacle?
>
> While our algorithms can be extended to include constraints through soft penalties or regularization in practical settings, recovering theoretical guarantees on convergence while guaranteeing hard constraint satisfaction at all times is a challenging open problem. Specifically, there are two key challenges: (i) Our problem involves deterministic dynamics, which is common in most control problems. Unlike stochastic environments where the analysis of policy gradient methods is simplified by assumption of suitable ergodicity and irreducibility properties of the underlying Markov decision process, proving well-definedness of the cost under constrained policies for deterministic systems will require establishing instead of simply assuming these ergodicity and irreducibility properties, which is challenging in general. (ii) The addition of constraints further complicates this analysis, as it can potentially disconnect the set of reachable states and render the objective function ill-defined, requiring careful analysis.
>
> While such problems are beyond the scope of this work, we will include a brief discussion on these challenges in the final version.
>
> **c) On visualization of reward curves:**
>
> >The visualization in the experimental section could be improved. For example, in Figure 1, it is recommended to truncate the curves to a smaller number of samples (e.g., the first 500) to better illustrate the dynamic changes. Currently, the mean trajectories are almost completely overlapped with the 1-SD bands, making it difficult to distinguish between different methods and limiting the readability and comparability of the results.
>
> **Response:** Thank you for the constructive feedback. We agree, and will truncate the first two plots appropriately so that it becomes easier to visualize the variations in both the subplots in the final version.
>
> **d) On experimental results in the main text:**
>
> >Since both PPO and LQR are included as baselines in the experimental section, it would be better to move the comparison with LQR into the main text to facilitate a clearer comparison of the results.
>
> >If possible, the stability experiments related to Figure 6 would be better presented in the main text to more intuitively demonstrate the results. Currently, the experimental section takes up less than one page and feels somewhat flat.
>
> **Response:** Thank you for this suggestion. We agree - while we presented the LQR and stability experiments in the Appendix due to space limitations, given that the final version allows for an additional page, we will move these results to the main text to facilitate ease of comparison, make the experiment section more self-contained, and strengthen the demonstration of the proposed algorithms.
>
> **e) Correction to Equation 3:**
>
> >Second equation in equation (3) should be:
> $g_{t+1} = \bar{A}g_t + \bar{B}u_t$
>
> **Response:** Thank you for pointing out this error. We will fix this in the final version.
>
> We hope that we have been able to address all your concerns. We look forward to any additional questions you may have, and hope that you will consider raising your evaluation score of our work accordingly.

---

> > ### Comment · Reviewer_1Cp5 · 2025-08-05
> >
> > Thank you for the detailed and well-structured rebuttal. Your clarifications on the assumptions were helpful and showed how they align with standard practices or can be relaxed without undermining the theory or performance. I also appreciate the insight you provided regarding constrained optimization—it’s good to see the authors aware of these challenges even if they lie beyond the current scope.
> >
> > I’m glad to hear that the visualization and formatting issues will be addressed in the final version. Moving the LQR and stability experiments into the main text will certainly enhance the self-containment and strength of the empirical section.
> >
> > Overall, I found the response thoughtful and convincing. I look forward to seeing the improved version of this paper.

---

> > > ### Author Response · Authors · 2025-08-05
> > >
> > > Thank you very much for taking the time to engage with our response! We appreciate your positive evaluation of our work and are glad that the clarifications on our assumptions and extensions to constrained settings were helpful. We look forward to incorporating your feedback on the presentation of the experimental section, which will certainly strengthen the paper.

---

### Official Review · Reviewer_fNRn · 2025-07-02

**Clarity:** 3
**Significance:** 2
**Originality:** 3
**Rating:** 4
**Confidence:** 4

**Summary:**

This paper presents a control approach that optimizes PID parameters using gradient descent (PG) methods. The problem is first formulated as an LQR problem; however, rather than adopting PG methods commonly used for LQR, the study directly computes the derivatives of each variable. This leads to a thorough mathematical derivation and establishes theoretical guarantees for convergence.

**Questions:**

In the understanding of model-free methods, sigma (σ) and the Gaussian mechanism indeed play crucial roles. As discussed in Section 3.2, the importance of the stochastic policy π(·|g) becomes evident, particularly in ensuring that π̃ₖ(u|g) > 0 holds for all states g and actions u.

1. "This suggests that in certain situations, introducing randomness is necessary in order to avoid issues that may arise from deterministic controllers μ̃ₖ(g)." While this statement highlights the necessity of randomness in avoiding problems associated with deterministic controllers. Does the presence of randomness also play a fundamental role in enabling the existence and effectiveness of policy gradients?
2. Why is the deterministic controller μ̃ₖ(g) unable to satisfy the condition π̃ₖ(u|g) > 0 when σ approaches zero? What are the theoretical implications of this limitation for the well-posedness of the policy gradient formulation?
3. How does this limitation impact the solution of the cost minimization problem within the framework of model-free reinforcement learning? Furthermore, in practical implementations, how should the value of σ be tuned to strike an appropriate balance between exploration and control performance?

**Ethical Concerns:**

["NO or VERY MINOR ethics concerns only"]

**Limitations:**

Despite the integration of reinforcement learning with PID control, the overall approach is limited by the inherent structural simplicity of PID controllers. As a result, the range of control problems that can be effectively addressed is relatively narrow.

**Quality:**

3

**Strengths And Weaknesses:**

Strengths:

The article addresses a clearly defined problem with a well-structured and logical approach. It discusses the issue that although the gain matrix K is fully determined, there may not always exist a feasible decomposition with respect to K. Instead of directly applying the PG method commonly used in LQR [17], the authors compute the gradients directly for the PID control problem. The paper progresses at a clear and efficient pace, addressing both model-based and model-free scenarios. Furthermore, by omitting certain differential terms, the method achieves a balance between theoretical soundness and practical applicability.

Weaknesses:

Although the paper integrates reinforcement learning with PID control, the applicability of the method is inherently limited due to the simple structure of PID controllers, which restricts their effectiveness to a narrow range of control problems.
The derivation is based on linear systems, and the paper does not address or discuss the extension to nonlinear systems.

---

> ### Author Rebuttal · Authors · 2025-07-30
>
> Thank you for your time and efforts in reviewing our work and for the positive evaluation of our contributions.  We address your individual concerns below.
>
> **a) On the applicability of PID controllers:**
>
> >Although the paper integrates reinforcement learning with PID control, the applicability of the method is inherently limited due to the simple structure of PID controllers, which restricts their effectiveness to a narrow range of control problems.
>
> **Response:** Thank you for allowing us to further clarify the novelty and applicability of our proposed approaches.
>
> To the best of our knowledge, there is no existing literature on policy gradient algorithms for PID control with ***provable theoretical guarantees***. While PID controllers have a simple parametric structure, we would like to emphasize this is a ***large and very useful class of control policies*** that are nearly ubiquitous in a variety of real-world industrial applications (chemical process control, robotics, aerospace). In fact, **>90% of controllers in industry employ PID architectures**, despite lack of theoretical guarantees [1].  The PID policy also offers built-in structural advantages including enhanced robustness as compared to LQR/variants and a low-dimensional parameterization that allows us to develop gradient-based algorithms with fast convergence properties. Yet, PID policies have received limited attention in the RL community. We also demonstrate these advantages in our benchmark experiments comparing the performance of our proposed designs with LQR (Figures 5 & 6) and PPO policies (Figure 1).
>
> Therefore, we believe that obtaining provably optimal and convergent policy gradient algorithms for the PID parameterization is **important and useful in a wide variety of applications**.
>
> [1] Hägglund, T. and Guzmán, J.L., 2024. Give us PID controllers and we can control the world. IFAC-PapersOnLine, 58(7), pp.103-108.
>
> **b) On extension to nonlinear systems:**
>
> >The derivation is based on linear systems, and the paper does not address or discuss the extension to nonlinear systems.
>
> **Response:** Thank you for bringing up this direction. We acknowledge and will add this consideration to our discussion in the final version. In this paper, we focused on linear systems with quadratic costs to obtain theoretical guarantees on global optimality and convergence for PID parameterized policies that is lacking in the existing literature. However, extension to nonlinear systems is an interesting yet challenging problem. Specifically, we believe that our model-free algorithm is particularly promising for this extension, since the gradients do not depend on the model parameters. Then, the challenge will be to establish gradient dominance, convergence and sample complexity guarantees with the underlying nonlinear dynamics, which will be the subject of future work.
>
> **c) On the stochastic policy in the model-free PG4PI algorithm:**
>
> We address the following questions jointly.
>
> >"This suggests that in certain situations, introducing randomness is necessary in order to avoid issues that may arise from deterministic controllers μ̃ₖ(g)." While this statement highlights the necessity of randomness in avoiding problems associated with deterministic controllers. Does the presence of randomness also play a fundamental role in enabling the existence and effectiveness of policy gradients?
>
> >Why is the deterministic controller μ̃ₖ(g) unable to satisfy the condition π̃ₖ(u|g) > 0 when σ approaches zero?
>
> **Response:** Indeed, building a stochastic policy centered around the deterministic controller **plays a fundamental role** in allowing us to compute explicit policy gradient expressions (line 215) and employ the policy gradient theorem to develop a model-free PG algorithm. In fact, this is **a key novelty of our approach**.
>
> To see why the **deterministic controller cannot satisfy the condition** $\pi(u|g) > 0$, we observe that the main technical challenge is that gradient in (16) needs to be well-defined in order to apply the policy gradient theorem to obtain a model-free algorithm. For a deterministic policy $\mu_{\tilde{K}}(g)$, the probability density $\pi(u|g) = \delta(u - \mu_{\tilde{K}}(g))$ is zero for all $u \neq \mu_{\tilde{K}}(g)$, making $\log \pi(u|g) = -\infty$ and its gradient $\nabla \log \pi(u|g)$ undefined almost everywhere. By introducing the stochastic policy (15) centered around the deterministic policy, we can ensure  $\pi(u|g) > 0$ for all $u, g$, making (16) well-defined, thus allowing us to apply the policy gradient theorem.
>
> >What are the theoretical implications of this limitation for the well-posedness of the policy gradient formulation?
>
> **Response:** As discussed above, we emphasize that the "stochastification" of the policy ensures $\pi(u|g) > 0$ for all $u, g$ is not a limitation, but rather **essential to ensuring well-posedness** of the formulation. In fact, the stochastic policy is necessary to guarantee that the policy gradients with respect to the PID parameters are well-defined, enabling us to obtain score functions (Theorem 3.2) and consequently the policy gradient estimates in a purely model free manner (Corollary 3.3) and establish a model-free PG algorithm for PID control.
>
> >How does this limitation impact the solution of the cost minimization problem within the framework of model-free reinforcement learning?
>
> **Response:** There is **no limitation with respect to cost minimization** and the solution is not affected by this choice of stochastic policy. In fact, we **theoretically guarantee** that the model-free algorithm PG4PI provably converges to the **globally optimal solution** (Theorem 4.7). This is once again enabled by the fact that we obtain exact gradient expressions with respect to our policy parameters (Corollary 3.3) and establish the gradient dominance properties  of the problem (Theorem 4.4) with the stochastic policy centered around the deterministic controller.
>
> **d) On the choice of $\sigma$:**
> >In practical implementations, how should the value of σ be tuned to strike an appropriate balance between exploration and control performance?
>
> **Response:** Thank you for this question on the tradeoffs involved in choosing the variance $\sigma$. From our ablation studies (Fig. 4), we find that varying the $\sigma$ does not have a significant impact on algorithm performance for small values ($\sigma$ < 1.0); however, some performance degradation occurs for larger variances ($\sigma=3$). Thus, in practical implementations, the performance and convergence profile of the algorithm can be maintained for a wide range of $\sigma$, allowing for choice based on other factors such as exploration and numerical stability.
>
> We hope that we have been able to address all your concerns through the above responses. We look forward to any additional questions you may have, and hope that you will consider raising your evaluation score of our work accordingly.

---

> ### Comment · Reviewer_fNRn · 2025-08-02
>
> This rebuttal provides a detailed and in-depth response to the reviewers' comments, demonstrating the authors' deep understanding of their research and a clear articulation of the strengths of their proposed method. Below is an evaluation of each section:
>
> a) Regarding the applicability of PID controllers, the authors emphasize the widespread use and importance of PID controllers in practical industrial applications, and point out the lack of existing literature on PID control policy optimization algorithms with theoretical guarantees. Nonetheless, this work remains confined to the scope of problems that PID control can address.
>
> b) Concerning the extension to nonlinear systems, the authors acknowledge that the current work focuses on linear systems, which are theoretically more tractable, and clearly state that extending to nonlinear systems is a potential direction for future work. This candid attitude and forward-looking perspective are commendable, although no direct results are currently presented in this regard.
>
> c) The explanation regarding the role of stochastic policies in the model-free PG4PI algorithm is thorough. The authors acknowledge that introducing randomness is essential for ensuring the validity of gradient computations. Indeed, this is not only a technical key point but also a key innovation of the study.
>
> d) On the choice of the parameter σ, the authors provide practical guidance based on experimental data, showing performance variations under different σ values. This is instructive for parameter tuning in practical applications.
>
> Overall, this rebuttal effectively addresses the reviewers' concerns and enhances the persuasiveness of the original manuscript through reference and theoretical analysis.

---

> > ### Author Response · Authors · 2025-08-03
> >
> > Thank you very much for taking the time to engage with our response! We appreciate your positive assessment of our rebuttal, and we’re glad that we could address your concerns and further clarify the technical contributions and significance of our work. We’re happy to address any additional questions you may have and hope that you will consider adjusting your final evaluation accordingly!

---

### Official Review · Reviewer_6Xsd · 2025-07-03

**Clarity:** 3
**Significance:** 3
**Originality:** 2
**Rating:** 4
**Confidence:** 3

**Summary:**

This paper proposes two algorithms (model-based and model-free, respectively) to integrate classic PID control within the RL framework, enabling the automatic learning the PID parameters. They do so using RL with theoretical guarantees of convergence and global optimality.

The proposal aims to bridging classic control theory with RL while retaining the practical robustness of PID controllers.

Experiments are performed in the Chemical Reactor and a 48-dimensional state-space LA University Hospital environments, showing that their proposal converges faster than a PPO, as it requires less exploration.

**Questions:**

- The authors acknowledge that further work is needed to extending the mathematical framework to encompass multi-objective trade-offs. Can they offer a brief explanation as to how they think it can be done?
- Will this approach be always preferable to the usual heuristic tuning of PID? If not always, when not?
- What is the rationale for choosing those two particular testing environments? They seem to be relative easy to solve.
- Is there any type of overhead (reserouces/time/extra data/....) required for PG4PID and PG4PI? If so, could you quantify them for your examples?
- Is this proposal applicable to more complex real-world systems as it is? Or some prerrequisites or preprocessing are needed? Will they work or are there any limitations to be aware of?

**Ethical Concerns:**

["NO or VERY MINOR ethics concerns only"]

**Final Justification:**

I acknowledge the effort from the authors on preparing the rebuttal and offering clarification to the raised points. Responses were appropriate and relevant, and they also reflected a deep understanding of your work and its place in the SOTA. I maintain my rating as I think this is an interesting paper, backed with evidence and well-defended in the rebuttal. However, additional work/experiments could have been done to make it rounder and for it to merit a stronger acceptance.

**Limitations:**

Limitations are listed in the conclusions:
- no multiobjetive yet.
- no hard constraints on states and inputs.
however, authors do not anticipate how they think they could be addressed.

No potential negative societal impact is foreseen.

**Paper Formatting Concerns:**

No concerns.

**Quality:**

3

**Strengths And Weaknesses:**

STRENGTHS:
- It address the relevant problem of how to improve the learning of RL in scenarios where PID controllers already work very well.
- The manuscript provides algorithms for model-free and model-based.
- Proofs of convergence and global optimality guarantees are provided for the two algorithms.
- The experiments are well-described and the analysis backed by the results.

WEAKNESSES:
- While interesting, the proposal seems quite incremental, (though offering theoretical guarantees is excellent).
- Applicability outside problems that can be controlled with PID is unclear (i.e. not clear if all problems could be somehow modeled as PID-controlled, and let the algorithms find the parameters, even though it would be hard for a human).
- The assumptions about the structural conditions make the proposal very concrete (though the amount of PID controllers out there is huge).

---

> ### Author Rebuttal · Authors · 2025-07-30
>
> Thank you for your time and efforts in reviewing our work, and for the positive evaluation of the theoretical guarantees in our work. We address your questions below.
>
> **a) On the novelty of our contribution:**
>
> >While interesting, the proposal seems quite incremental, (though offering theoretical guarantees is excellent). Applicability outside problems that can be controlled with PID is unclear (i.e. not clear if all problems could be somehow modeled as PID-controlled, and let the algorithms find the parameters, even though it would be hard for a human).
> >The assumptions about the structural conditions make the proposal very concrete (though the amount of PID controllers out there is huge).
>
> **Response:** Thank you for allowing us to further clarify the novelty and applicability of our proposed approaches. To the best of our knowledge, there is no existing literature on policy gradient algorithms for PID control with ***provable theoretical guarantees***. While we do not claim that all problems could be controlled by PID controllers, we would like to emphasize this is a **large and very useful class of control policies** that is nearly ubiquitous in a variety of real-world industrial applications (chemical process control, robotics, aerospace), as the reviewer also rightly points out. In fact, **>90% of controllers in industry employ PID architectures**, despite lack of theoretical guarantees [1].  The PID policy also offers built-in structural advantages including enhanced robustness as compared to LQR/variants and a low-dimensional parameterization that allows us to develop gradient-based algorithms with fast convergence properties. Yet, PID policies have received limited attention in the RL community. We also demonstrate these advantages in our benchmark experiments comparing the performance of our proposed designs with LQR (Figures 5 & 6) and PPO policies (Figure 1).
>
> We would also like to take this opportunity to highlight that obtaining theoretical guarantees on global optimality for this important class of problems is highly challenging. We derive novel policy gradient expressions for the PID control problem in both the model-based and model-free settings (Theorems 3.2 & 4.1, Corollary 4.2). We also establish gradient dominance of this problem (Theorem 3.3), which allows us to provide proofs of optimality, convergence, and sample complexity (Theorems 5.1, 5.3, & 5.6). In the model-free setting, we propose a novel policy gradient approach (Algorithm 2) where the key innovation stems from the fact that we are provably learning the optimal parameters $\tilde{K}^*$ of a deterministic PID controller while updating the stochastic policy of eq. (15), wherein: (i) the PID gradients derived in Sec. 4.2 are new and necessary, and (ii) the guarantees of Sec. 5.2 require substantial innovation over existing analyses.
>
> Taken together, these are nontrivial, novel contributions that extend policy gradient algorithms to the widely useful class of PID parameterized policies.
>
> [1] Hägglund, T. and Guzmán, J.L., 2024. Give us PID controllers and we can control the world. IFAC-PapersOnLine, 58(7), pp.103-108.
>
> **b) On extension to multi-objective settings:**
>
> >The authors acknowledge that further work is needed to extending the mathematical framework to encompass multi-objective trade-offs. Can they offer a brief explanation as to how they think it can be done?
>
> **Response:** Thank you for raising this question. This extension is a subject of future work, as obtaining theoretical guarantees in multi-objective optimization in general, and multi-objective RL in particular is a challenging problem. Specifically, building on works such as [2-4], we can address this problem by scalarization (linearly combining multi-objective trade-offs to obtain a single-objective problem). The key challenge would be to derive policy gradient expressions and establish gradient dominance for this new objective with PID parameterized policies, to obtain policy gradient algorithms with theoretical guarantees to achieve pareto-optimality.
>
> [2] K. Van Moffaert, M. M. Drugan and A. Nowé, "Scalarized multi-objective reinforcement learning: Novel design techniques," 2013 IEEE Symposium on Adaptive Dynamic Programming and Reinforcement Learning (ADPRL), Singapore, 2013, pp. 191-199
>
> [3] Van Moffaert, K. and Nowé, A., 2014. Multi-objective reinforcement learning using sets of pareto dominating policies. The Journal of Machine Learning Research, 15(1), pp.3483-3512.
>
> [4] Petchrompo, S., Coit, D.W., Brintrup, A., Wannakrairot, A. and Parlikad, A.K., 2022. A review of Pareto pruning methods for multi-objective optimization. Computers & Industrial Engineering, 167, p.108022.
>
> **c) Comparison to heuristic tuning:**
>
> >Will this approach be always preferable to the usual heuristic tuning of PID? If not always, when not?
>
> **Response:** Thank you for raising this question. Our policy gradient PID approach is preferable to heuristic tuning, for two reasons:
>
> (i) We derive ***exact analytical expressions for the policy gradients*** in terms of the policy parameters from Theorem 3.1 and Corollary 3.3, which allows for **fast convergence** of our algorithms even compared to other gradient-based tuning methods like PPO (see experiments in Section 5). In contrast, **heuristic tuning involves trial-and-error** and searching over a large non-convex space of policy parameters that will yield slower convergence.
> (ii) Our algorithms are accompanied by **theoretical proofs** of convergence to **globally optimal solutions**. In contrast,  **heuristic tuning is highly likely to yield local minima** due to the non-convex landscape of the underlying optimization problem.
>
> Given the additional page allowed for the final version, we will add an additional benchmark experiment demonstrating this limitation of heuristic training with respect to our algorithms.
>
> **d) On choice of environments:**
>
> >What is the rationale for choosing those two particular testing environments? They seem to be relative easy to solve.
>
> **Response:** We chose standard ControlGym environments to demonstrate our proposed approach. We would like to highlight that the environments have high-dimensional state spaces (8- and 48- dimensional state spaces respectively for the chemical reactor and LA hospital environments) that pose a challenge to classical design approaches. In fact, in our experiments, PPO required a large number of samples (nearly 30000, see Fig. 1) for training with no proof of optimality and LQR was shown to be non-robust under dynamical errors in these environments (Fig. 6), while our algorithms demonstrate fast convergence in only a few samples (Fig. 1). Finally, we note that environments such as a chemical reactor and building control (LA Hospital) represent important applications for PID control in the industry, further motivating the choice of environments.
>
> **e) On resource overhead:**
>
> >Is there any type of overhead (reserouces/time/extra data/....) required for PG4PID and PG4PI? If so, could you quantify them for your examples?
>
> **Response:** Since our proposed method has a structured policy parametrization and explicit gradient expressions, the simulations only needed modest computational power; specifically, we ran our experiments on a a 6-core i7-8750H, 2.20GHz CPU, an NVIDIA GeForce RTX 2070 GPU, and 32GB RAM (line 759). We do not require any additional resource overhead or data beyond those detailed in Appendix A.4.
>
> **f) On applicability to real-world problems:**
>
> >Is this proposal applicable to more complex real-world systems as it is? Or some prerrequisites or preprocessing are needed? Will they work or are there any limitations to be aware of?
>
> **Response:** We believe that our proposed method can be used to solve a wide class of real-world control problems.
> (i) First, as we discussed in our earlier response, PID control is applicable to a wide variety of complex real-world applications, including aviation, robotics, power plants, chemical plants etc.
> (ii) We do not need preprocessing of any of the components in our proposed method; rather, we directly compute exact policy gradient expressions and utilize them in our policy updates.
> (iii) As mentioned in the Conclusion of our paper, we propose to extend this approach to multi-objective and constrained settings in the future, which will further expand the applicability to real-world settings.
>
> **g) On constrained design:**
>
> >no hard constraints on states and inputs. however, authors do not anticipate how they think they could be addressed.
>
> **Response:** While our algorithms can be extended to include constraints through soft penalties or regularization in practical settings, recovering theoretical guarantees on convergence while guaranteeing hard constraint satisfaction at all times is a challenging open problem. Specifically, there are two key challenges: (i) Our problem involves deterministic dynamics, which is common in most control problems. Unlike stochastic environments where the analysis of policy gradient methods is simplified by assumption of suitable ergodicity and irreducibility properties of the underlying Markov decision process, proving well-definedness of the cost under constrained policies for deterministic systems will require establishing instead of simply assuming these ergodicity and irreducibility properties, which is challenging in general. (ii) The addition of constraints further complicates this analysis, as it can potentially disconnect the set of reachable states and render the objective function ill-defined, requiring careful analysis. While such problems are beyond the scope of this work, we will include a brief discussion on these challenges in the final version.
>
> We hope that we have been able to address your concerns through the above responses. We look forward to any additional questions you may have, and hope that you will consider raising your evaluation score of our work accordingly.

---

> > ### Comment · Reviewer_6Xsd · 2025-08-05
> >
> > Thank you for the hard work you put into preparing the rebuttal and into offering clarification to the raised points.
> > I acknowledge that your responses to my questions and concerns—as well as those raised by my fellow reviewers—are appropriate and relevant. They also reflect a deep understanding of your work and its place in the SOTA.
> > I maintain my rating as I think this is an interesting paper, backed with evidence and well-defended in the rebuttal. However, additional work/experiments could have been done to make it rounder and for it to merit a stronger acceptance.

---

> > > ### Author Response · Authors · 2025-08-05
> > >
> > > Thank you for taking the time to engage with our response and for your constructive feedback! We’re glad that our clarifications were helpful and are happy to answer any further questions you may have.

---

### Decision · Program_Chairs · 2025-09-17

**Decision:**

Accept (poster)

**Comment:**

The authors study policy-gradient methods for PID-based policies. The scope is thus narrow, but of great practical relevance.
As convincingly reiterated by the authors during the discussion, this work provides the first theoretical guarantees for this specific setting. The value of this contribution outweights the remaining concerns of the reviewers on the limited scope and experiments.
The reviewers' suggestions should be included in the final version.